

1  **FRESHWATER FISH FAUNA OF RIVERS OF**
2  **SOUTHERN WESTERN GHATS, INDIA**

Anbu Aravazhi Arunkumar[1], Arunachalam Manimekalan[2]

[1]Department of Biotechnology, Karpagam Academy of Higher Education, Coimbatore 641 021.

Tamil Nadu, India

[2]Department of Environmental Sciences, Biodiversity and DNA Barcoding Lab, Bharathiar University,

Coimbatore 641 046, Tamil Nadu, India.

*Correspondence to*: Anbu Aravazhi Arunkumar (anbu.arunkumar@gmail.com)
https://doi.org/10.1594/PANGAEA.882214

**Abstract.** We studied the freshwater fish fauna of Rivers of Southern Western Ghats for a period of three years

from 2010 to 2013. We recorded 64 species belonging to 6 orders, 14 families and 31 genera. Alteration in the

micro and macro habitats in the system severely affects the aquatic life especially fishes and also complicates

the fish taxonomy. In the present study a total of 31 sites of six river systems of Southern Western Ghats were

studied in which a total of 64 species belonging to 6 orders, 14 families and 31 genera were recorded. Among

the 64 species *Cyprinidae* was the dominant family with 3 family 18 genus and 49 species (76.6%) compared to

other order and families, further the data analyses suggested that species belonging to the order Cypriniformes

were found to be the dominant species in the locations considered in the present survey. Interestingly, among the

31 sites Thunakadavu stream, Gulithuraipatti, Athirappalli, Naduthotam, Nadathittu, Mullaithodu,

Thonanthikla, Noolpuzha and Sinnaru exhibited high variations in species abundance and as well species

richness. Fifteen out of the 64 fish species endangered to the Western Ghats. *Garra periyarensis* and *Cirrhinus*

*cirrhosus* are known to be vulnerable and *Hemibagrus punctatus* is Critically Endangered because of various

anthropogenic activities. The significances of the study and timely measures needed to protect the species have

also been concisely discussed.

**Keywords:** Southern Western Ghats, Water Quality, Species Diversity, Endemics, threats, Conservation.

**1. INTRODUCTION**

The Western Ghats of India has a rich freshwater fish fauna with a high level of endemism (Dahanukar

*et al.,* 2004). However, current knowledge of the threats faced by Western Ghats fishes suggests that a major

part of this fauna is threatened by human activities and invasive alien fish species (Dahanukar *et al.,* 2004).

Thus, knowledge of the diversity and distribution of the fish fauna is essential for designing and implementing

conservation strategies. However, data on the fish fauna of the Western Ghats have limitations as most of the

rivers have not been surveyed extensively and checklists for individual rivers are not available. In the present

study we document the freshwater fish fauna of the the long and meandering eastward flowing river systems of



Southern Western Ghats, especially from Bhavani River System, Moyar River System, Chalakudy River System, Periyar River System, Cauvery River System and Nugu River System, in the southern region of the Western Ghats.

History of the Indian freshwater fishes is way back to Hamilton (1822) on the fishes found in the river Ganges and its tributaries. The documentation and listing of the fishes from different part of India was carried out mainly by Jerdon (1848). A comprehensive and authoritative account on the freshwater fishes has been provided by Day (1865 – 1878). The further investigations on the freshwater fishes of India especially the Western Ghats was initiated by Hora (1921; 1937; 1938; 1941; 1942; 1949) and he enunciated the Satpura Hypothesis. These led to the new descriptions, enlisting with elaborate discussions on the endemism and other zoogeographical relevance and several new taxa have been added from Kerala during this period.

Studies on the endemic fishes from various streams and rivers in the Western Ghats mountain ranges have been compiled. Fish diversity in selected streams in northern Karnataka (Arunachalam *et al,*. 1997); Central Western Ghats (Arunachalam 2000) have been reported. Arunachalam *et al.,*(2005) reported a new fish species *Neolissocheilus wynaadensis* from the Karnataka part of Western Ghats. Arunachalam (2007) have reported *Psilorhynchus amplicephalus*, a new species from Balishwar river of Assam, India. Earlier Biju *et al.,* (1996) has recorded *Puntius filamentous* (Val.) and *Puntius melanampyx* (Day) in Orukomban and Thelikal during the survey from December 1996 to May 1997.  Manimekalan (2002) has rediscovered the critically endangered air birthing cat fish *Clarias dayi* hora (Pisces:*Claridae)* from Mudumalai Wildlife Sanctuary. Manimekalan (1998) has described a new species *Glyptothorax davissinghi Manimekalan and das* (Pisces: *Sisoridae*), a new cat fish from Nilambur in the Nilgiri Biosphere, South India. Manimekalan (1997) made a new recorded of *Schismatorhynchus (Nukta) nukta* (Sykes) (Pisces: Cyprinidae) from Moyar river. Arunkumkar *et al*., (2015) has recorded 37 species from Cauvery river system. Silas (1951) listed 25 fish species from Anamalai hills and 10 species from Neliampathi hills. His study extended the distribution of several species earlier known only from the central division of the Western Ghats to the southern division beyond the Palghat gap.

## 2. METHODOLOGY

### 2.1 Collection and Identification

Fishes were collected using cast net, dip net, gill net and drag net from various streams and rivers of Southern Western Ghats. At most care was taken not to damage the species while collecting. A total of 5 specimens from each species were collected and fishes were photographed before it was preserved in formalin so that the fishes can be photographed with original colour. Further the specimens were preserved in 10 per cent formalin for smaller samples and for larger samples formalin has been injected into the abdominal cavity so that the internal organs are well preserved for further taxonomic studies. The specimens were tagged and the reference numbers were given for specimen identification and transported to Lab. The species were identified based on the key given by Talwar and Jhingran (1991),  Jayaram (1999 & 2010) and Menon (1992). Holotype and paratypes of species were examined in Zoological Survey of India, Southern Regional station, Chennai and 2olkata for confirmation of species. Voucher specimens have been made for each species and deposited at the Biodiversity and DNA Barcoding Lab, Dept. of Environmental Sciences, Bharathiar University.



**2.2 Physico-chemical Analysis of the Water Quality at Sampling Sites**

Water samples were collected from all the seven sampling stations during post-monsoon, the depth of 10cm. Water quality analyses such as pH, conductivity, turbidity, total dissolved solids (TDS), resistivity, salinity, dissolved oxygen (DO), and water temperature were done as per the regulations of APHA 1995, respectively. Field analysis of the samples was done using portable water analyzer (X tech, Nagman Instruments Electronics, India) (Gurumurthy and Tripti, 2015; Thomas *et al.*, 2015; Anushiya and Ramachandran, 2015).

**2.3 Interpretative analysis**

To quantify species diversity, for the purposes of comparison, a number of indices have been followed. To measure the species diversity (H) the most widely used Shannon index (Shannon and Weaver, 1949), Evenness index (E) (Pielou, 1975), and Dominance index (D) (Simpson, 1949) were used. Similarity coefficients of the fish community were calculated by using the widely used Jaccard index (Southwood, 1978). The above statistical analyses were performed using SPSS software.

**2.4 Data processing and analysis**

Further, the data from different appropriate sources are coded and recorded into a database system. For the accuracy of the data recorded at every source of the survey, correspondence between elementary data sheets and the original coding sheets were considered; accuracy and quality of the data were inspected up, edited, and coded at the field level.

**3. RESULTS AND DISCUSSION**

Fish Fauna were surveyed from the streams and rivers of Southern Western Ghats. Collection sites were selected based on the earlier faunal distribution published in literature. The Western Ghats is a mountain range that runs almost parallel to the western coast of Indian peninsula. It is a UNESCO World Heritage Site and is one of the eight "hottest hotspots" of biological diversity in the world. It is also called as "The Great Escarpment of India". The range of Western Ghats runs from north to south along the western edge of the Deccan Plateau, and separates the plateau from a narrow coastal plain, called Konkan, along the Arabian Sea. A total of thirty nine world heritage sites including national parks, wildlife sanctuaries and reserve forests - twenty in Kerala, ten in Karnataka, five in Tamil Nadu and four in Maharashtra adds fame to the Western Ghats. Fish fauna were collected from the long and meandering eastward flowing river systems of Southern Western Ghats, especially from Bhavani River System, Moyar River System, Chalakudy River System, Periyar River System, Cauvery River System and Nugu River System. The study sites and its characteristics are recorded and presented in Table 1 and Fig 1. In the present study a total of 31 sites of six river systems of Southern Western Ghats were studied in which a total of 64 species belonging to 6 orders, 14 families and 31 genera were recorded (Table. 2). Among the 64 species *Cyprinidae* was the dominant family with 3 family 18 genus and 49 species (76.6%) compared to other order and families (Fig.2, Fig.7).



### 3.1 Fish Species Density, Abundance, and Distribution

Among the 31 sites high species diversity was recorded at Sinnaru of Cauvery River system (H'- 1.268) and low diversity was recorded at Thunakadavu tunnel, Chalakudy River System recorded (H'- 0.357) (Table: 3, Fig: 3). The maximum species richness was recorded in Sinnaru (S – 21) and the minimum species richness was recorded at Puliyarkutti 3[rd] bridge, Thunakadavu tunnel and Sorrakottaodai (S – 3), (Table: 3, Fig: 4). The maximum species abundance 152 was recorded at Naduthottam and lowest abundance 16 was recorded at Sorrakottaodai and Belikoondu (Table: 3, Fig: 5). The maximum dominance (D - 21.346) was recorded at Sinnaru and lowest dominance (D- 2.121) was recorded at Thunakadavu tunnel (Table: 3).

### 3.2 Species composition

Species similarity between the sites was very less among 31 sites of six river systems. Cluster analysis showed that similar species composition between the sites based on the species diversity. (Table:4, Fig: 6). Totally 5 clusters were grouped for 31 sites of six river systems of southern Western Ghats from which it's clearly seen that most of the sampling sites were clustered together because of the similarity of species composition among the sites. Several sites where human disturbances are prevalent also fall in the same cluster. Certain sites remain separate, because only species composition in that particular site is not present in the other location. There are two main reasons for this separate clustering – 1. due to the rare species forms and 2. due to low water temperature.

### 3.3 Water Quality:

Water Quality parameters were recorded and presented in table 2.6. It is found that the parameters value lies between the IS: 10500 Permissible limits. (Table: 6). The acidic or alkaline nature of the water will be decided based on the pH level. Water pH ranges between 6.5 to 8.5, Kadapilliyarthittu (pH - 9) was recorded with pH level is high and Anjurily, Athirapalli, Urilikal (pH – 7.2) recorded low pH level compared to the other sites. Low conductivity value 27.8mS was recorded in Puliyarkutti river 8[th] bridge and Puliyarkutti river 3[rd] bridge sites and high conductivity value 85.2mS recorded in Noolpuzha of Nugu river system. Total dissolved solids (TDS) are a measure of inorganic salts dissolved in water. This dissolved solid comes from both natural and human sources. Mitchell and Stapp in 1992 have suggested Changes in TDS concentrations that can be harmful. If TDS concentrations are too high or too low, the growth of much aquatic life can be limited, and death may occur. Thenkasithodu witnessed a low value of TDS content as 13.7 mg/l and Urilikal recorded a high value of TDS as 51.9mg/l. A minimum Resistivity value of 2.58 was measured at Kadapilliyarthittu and a maximum 45.6 was measured at Thenkasithodu. A high level of DO was recorded at Thenkasithodu as 6.11mg/l and a low level of DO was recorded at Belikoondu 0.63 mg/l. Arunkumar *et* al., (2015) recommended that the lowest DO recorded at sampling sites is due to organic-rich domestic waste let into the river by the tourists in the river system. Low value of salinity was recorded at sites viz., Thenkasithodu, Anjurily, Sorrakottaodai, Naduthotam, Nellithurai, Kovaikutralam falls, Puliyarkutti River 8[th] bridge and Puliyarkutti River 3[rd] bridge as 0.01 ppt and a high level of salinity was noted at Kadapilliyarthittu as 0.18ppt. Maximum water temperature was recorded at Pillapara as 33.6°C and a minimum water temperature was noted at Thenkasithodu as 18.9°C.





Rajan (1955) has studied the fishes of Moyar river system and has reported 48 species. Manimekalan
(1998) has reported 38 species form Mudumalai wildlife sanctuary. Manimekalan has stated that species like
*Labeo dero, Puntius mudumaliensis, Schimatorhynchus nukta, Danio neilgherriensis, Crossochelius latius*
*latius, Clarias dayi, Gambusia affinis* were restricted to Moyar river system. Also *Clarias dayi* a critically
endangered species has been recorded by Manimekalan (2002). *Puntius carnaticus* and *Danio aequpinnatus* was
recorded as common species of Moyar river system. Rajan (1955) and Mukerjii (1931) has studied the
headwaters of Bhavani river and reported species like *Travancoria elangata, Barilius canarensis, Rasbora*
*caveri, Garra menoni, Silurus wynaadensis* were restricted to Bhavani River system. *Puntius filamentosus,*
*Puntius melanampyx, Puntius carnaticus, Barilius gatensis, Danio aequpinnatus, Rasbora daniconius* were very
common in Bhavani River System. Arunkumkar *et al*., (2015) has recorded 37 species from Cauvery river
system. Among several fish species recorded, the only *Garra gotyla stenorhynchus* is reordered as one of the
endangered species in Grand Anicut Cauvery, which is locally consumed (Murthy *et al*., 2015). But *Garra*
*gotyla stenorhynchus* is still under least concern status of IUCN.

Silas (1951) in his faunal account discusses the extension of range of *Salmostoma acinaces* (*Chela*
*argentea* Day), Barbodes carnaticus (*Barbus* (*Puntius*) *carnaticus*), Osteochilus (*Osteochilichthys*) *thomassi* and
*Batasio travancoria* and lists 2 endemic species described by Herre viz. *Homoloptera Montana* and
*Glyptothorax housei*. Silas further reported 5 species from the Cochin part of the anamalai hills viz. *Barilius*
*bakeri, Puntius denisoni, Travancoria jonesi, Noemacheilus triangularis* and *Batasio travancoria. Punitus*
*bimaculatus* earlier considered as a juvenile of *Puntius dorsalis* has been collected from these hills. Interestingly
this species is found to be the most dominant *Puntius* species in the hill ranges of the Eastern Ghats especially
Javadi hills. *Puntius punctatus* earlier considered as a synonym of *Punitus ticto* has been kept as a separate
species and both these species have been collected from Anamalai hills (Menon, 1999).

Diversity in the Anamalais is very high except for a few areas such as the Aliyar river basin. The lack
of diversity in the Aliyar river basin is due to the fact that most of the streams in the area are non-perennial and
are prone to disturbance/contamination by the local tribal people. This diversity is attributed to the controlled
fishing activity by locals and protection by Forest officials. The physical environment like forest vegetation,
riparian vegetation, water temperature, habitat type, and in-stream cover (which provide hiding places for fish)
play a major role in species diversity and richness.
Altitude also plays a major role in species diversity. Colinvaux (1930) proposed the theory of diversity
that changes with altitude on mountainsides – diversity is lowest at high elevation and vice versa. The present
finding supports the above theory. The westward flowing Periyar River originates near moolavaigae and reaches
the Periyar Lake. The Periyar Tiger Reserve is one of the biodiversity rich areas in southern Western Ghats from
where the Periyar River originates, (Silas 1950, 1952; Kurup *et al.,* 2004). Earliest studies on the fish fauna of
the PTR dates back to 1948 when Chacko (1948) listed 35 species from the Periyar Lake, including the critically
endangered small scaled *Schizothoracin Lepidopygopsis typus*. Later Menon & Remadevi (1995) described
Hypselobarbus *kurali* from streams adjoining the Periyar Lake raising the total number of fish species to 38. In
the present study 64 species were collected from 31 study sites of six river systems of Southern Western Ghats.



Species like *Puntius melanampyx, Puntius carnaticus, Puntius amphibious, Puntius fasciatus, Puntius mahecola, Devario aequipinnatus, Garra mullya, Travancoria jonesi, Nemacheilus guntheri* were commonly found in all the six river systems (Fig:7).

Smith has stated that habitat selection of the fishes is influenced by the body structure, food and shelter and by physiological process. Moreover the fish analyses the characters of the rivers and streams and further they respond to the characters and helps themselves for the survival of the fittest. Hence it is reliable that the Micro and Macro habitat plays a key role in the morphology and physiological characters and modifications of the species. The fish prefers the habitat based on the nature of the rivers or stream substratum type where the muddy bottom with debris is records for high species richness of the bottom feeders. Odum (1945) well stated that the flow of the water in the channel is an important factor prevailing the distribution of fishes, the species like *Barilius, Hypselobarbus, Puntius, Travancoria, Rasbora* and *Tor* prefers fast flow. The nature of the substratum and the flow rate seem to be more or less closely interrelated in governing the distribution of the fishes. This induces the dominance of the cyprinid species to be well flourished in all the river systems, of the Southern Western Ghats. It is clear that Ecological structure plays a key role in representing River Systems of Southern Western Ghats which is flourished with rich species diversity and abundance.

## 4. SUMMARY

The morphological-based fish taxonomy is more inconclusive because the micro and macro habitat have influenced the morphological variations within the species. In the present study, the fish species were collected by using different mesh size of gill nets, cast net and dip nets from the long and meandering eastward flowing river systems of Bhavani, Moyar, Chalakudy, Periyar, Cauvery and Kabini. In the present study a total of 31 sites of six river systems of Southern Western Ghats were studied in which a total of 64 species belonging to 6 orders, 14 families and 31 genera were recorded. Among the 64 species *Cyprinidae* was the dominant family with 3 family 18 genus and 49 species (76.6%) compared to other order and families, further the data analyses suggested that species belonging to the order Cypriniformes were found to be the dominant species in the locations considered in the present survey. Interestingly, among the 31 sites Thunakadavu stream, Gulithuraipatti, Athirappalli, Naduthotam, Nadathittu, Mullaithodu, Thonanthikla, Noolpuzha and Sinnaru exhibited high variations in species abundance and as well species richness. Importantly, the present study clearly documented that altitudes play a major role in species diversities and as well in species abundance. The fish is a healthy and high protein rich food, are in peril in Southern Western Ghats and the comprehensive listing of various species distribution and continuous monitoring is the most critical need of protection in the present scenario. It is very apparent to mention that the use of explosives, poisons and fishing of juveniles could be the primary causes to the sharp decline of the fish population in the study areas. Establishment of sanctuaries, preservation of genetic materials, awareness programmes and enforcement of laws are some of the short and long term remedial measures for the efficient conservation of faunal population in Southern Western Ghats. Social workers, fishermen and local people must also be educated about the importance of conservation of fish fauna in their area in general, so that the personnel in turn can also make awareness among the people in an ecological spirit.



**5.   ACKNOWLEDGEMENT**

The authors gratefully acknowledge facilities provided by the Department of Environmental Sciences,
Biodiversity and DNA Barcoding Laboratory, Bharathiar University.

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





**Table 1: Study site and its Habitat characteristics**

| S. No | Study site | Latitude and longitude | Altitude | Forest type | Stream order | Stream Width (m) | Stream Depth (m) | Area (m²) | Volume (m³) | Mean Velocity* (m/sec) |
|---|---|---|---|---|---|---|---|---|---|---|
| | | | | **Moyar River System** | | | | | | |
| 1 | Gulithuraipatti | 11° 36' N and 76° 47' E | 312 | Thorn forest | 4 | 10 | 6 | 1000 | 6000 | 4 |
| 2 | Kallampalayam | 11° 31' N and 77° 0' E | 300 | Thorn forest | 4 | 13 | 8 | 1300 | 10400 | 4 |
| 3 | Belemeenthurai | 11° 36' N and 76° 47' E | 520 | Dry deciduous | 4 | 19 | 1.75 | 1900 | 3325 | 4 |
| | | | | **Chalakudy River System** | | | | | | |
| 4 | Orukomban range | 10° 22' N and 76° 39' E | 450 | Dry deciduous | 4 | 6 | 0.5 | 600 | 300 | 3 |
| 5 | Thenmudiparai | 10° 24' N and 76° 36' E | 510 | Dry deciduous | 5 | 25 | 1.5 | 2500 | 3750 | 3 |
| 6 | Baghapallam | 10° 27' N and 76° 43' E | 748 | Dry deciduous | 5 | 8 | 0.5 | 800 | 400 | 3 |
| 7 | Thellikal | 10° 27' N and 76° 44' E | 840 | Dry deciduous | 4 | 4 | 1.0 | 400 | 400 | 3 |
| 8 | Puliyarkutti 8th bridge | 10° 23' N and 76° 40' E | 527 | Dry deciduous | 4 | 19.2 | 1.2 | 1920 | 2304 | 3 |
| 9 | Puliyarkutti 3rd bridge | 10° 23' N and 76° 41' E | 512 | Dry deciduous | 4 | 37 | 1.5 | 3700 | 5550 | 3 |
| 10 | Thunakadavu stream | 10° 25' N and 76° 46' E | 510 | Dry deciduous | 4 | 13.6 | 0.5 | 1360 | 680 | 3 |
| 11 | Thunakadavu tunnel | 10° 20' N and 76° 34' E | 520 | Dry deciduous | 5 | 15 | 10 | 1500 | 15000 | 5 |
| 12 | Urilikal | 10° 19' N and 76° 53' E | 3238 | Dry deciduous | 2 | 7 | 1.5 | 700 | 1050 | 2 |
| 13 | Athirappalli | 10° 18' E and 76° 34' N | 202 | Semi evergreen | 4 | 8 | 3 | 800 | 2400 | 4 |
| 14 | Pillapara | 11° 36' N and 76°47' E | 267 | Semi evergreen | 4 | 5 | 2 | 500 | 1000 | 4 |

| | | | | | | | | | |
|---|---|---|---|---|---|---|---|---|---|
| **Bhavani River System** | | | | | | | | | |
| 15 | Kovaikutralam falls | 10° 56' N and 76°41' E | Semi evergreen | 2 | 5 | 1.2 | 500 | 600 | 4 |
| 16 | Nellithurai | 11° 17' N and 76°53' E | Thorn forest | 4 | 27 | 1.1 | 2700 | 2970 | 5 |
| **Periyar River System** | | | | | | | | | |
| 17 | Oorpannikaham | 09° 28' N and 77°16' E | Evergreen | 4 | 12 | 2.1 | 1200 | 2520 | 2 |
| 18 | Valukuparai | 09° 28' N and 77°17' E | Evergreen | 4 | 7.5 | 0.3 | 750 | 225 | 3 |
| 19 | Melaparai | 09° 26' E and 77° 18' N | Evergreen | 4 | 11 | 4.2 | 1100 | 4620 | 3 |
| 20 | Naduthotam | 09° 26' N and 77° 19' E | Evergreen | 4 | 7.5 | 0.3 | 750 | 225 | 3 |
| 21 | Ummikuppamthodu | 09° 28' N and 77° 14' E | Evergreen | 4 | 5 | 3.0 | 500 | 1500 | 4 |
| 22 | Sorrakottaodai | 09° 28' N and 77° 15' E | Evergreen | 4 | 7 | 1.5 | 700 | 1050 | 3 |
| 23 | Mullaithodu | 09° 31' N and 77° 16' E | Evergreen | 4 | 10 | 0.6 | 1000 | 600 | 3 |
| 24 | Anjurily | 11° 36' N and 76°47' E | Evergreen | 4 | 20 | 5 | 2000 | 10000 | 2 |
| 25 | Thenkasithodu | 11° 36' N and 76°47' E | Evergreen | 4 | 11.3 | 0.5 | 1130 | 565 | 2 |
| **Cauvery River System** | | | | | | | | | |
| 26 | Kadapilliyarthittu | 12° 07' N and 77° 46' E | Dry deciduous | 4 | 75 | 1.5 | 7500 | 11250 | 2 |
| 27 | Belikoondu | 12° 11' N and 77° 43' E | Dry deciduous | 4 | 80 | 10 | 8000 | 80000 | 5 |
| 28 | Nadathittu | 12° 08' E and 77° 44' N | Dry deciduous | 4 | 70 | 6 | 7000 | 42000 | 3 |
| 29 | Sinnaru | 12° 06' N and 77° 46' E | Dry deciduous | 4 | 55 | 0.5 | 5500 | 2750 | 3 |
| 30 | Thonanthikla | 12° 07' N and 76° 46' E | Dry deciduous | 4 | 25 | 1 | 2500 | 2500 | 4 |
| **Nugu River System** | | | | | | | | | |
| 31 | Noolpuzha | 11° 41' N and 76° 23' E | Semi evergreen | 3 | 25 | 4.1 | 2500 | 10250 | 4 |

*Velocity (m/sec): 1. Very slow (<.05); 2. Slow (0.05-0.2); 3. Moderate (0.2-0.5); 4. Fast (0.5-1.0); 5. Very fast (>1).





**Table 2: List of Freshwater Fauna recorded during the present study**

| S.no | Species | Distribution locations | IUCN |
|---|---|---|---|
| | **Order: Cypriniformes** | | |
| | **Family: Cyprinidae** | | |
| | **Sub - Family: Cyprininae** | | |
| 1 | *Puntius melanampyx* | 18 | DD |
| 2 | *Puntius carnaticus* | 10 | LC |
| 3 | *Puntius amphibius* | 4 | DD |
| 4 | *Haludaria fasciatus* | 11 | LC |
| 5 | *Dawlinsia filamentosus* | 4 | LC |
| 6 | *Puntius sarana sarana* | 4 | LC |
| 7 | *Puntius dorsalis* | 2 | LC |
| 8 | *Puntius chola* | 2 | LC |
| 9 | *Puntius sophore* | 1 | LC |
| 10 | *Eechathalakenda ophicephalus* | 2 | EN |
| 11 | *Puntius mahecola* | 7 | DD |
| 12 | *Pethia conconius* | 4 | LC |
| 13 | *Sahyadria denisonii* | 2 | EN |
| 14 | *Sahyadria chalakudiensis* | 2 | EN |
| 15 | *Puntius sarana spirulus* | 1 | LC |
| 16 | *Puntius bimaculatus* | 3 | LC |
| 17 | *Pethia ticto* | 1 | LC |
| 18 | *Cirrhinus cirrhosus* | 2 | VU |
| 19 | *Skymatorynchus nukta* | 3 | EN |
| 20 | *Labeo boggut* | 1 | LC |
| 21 | *Labeo kontius* | 1 | LC |
| 22 | *Labeo ariza* | 3 | LC |
| 23 | *Labeo calbasu* | 2 | LC |
| 24 | *Labeo boga* | 2 | LC |



| | | | |
|---|---|---|---|
| 25 | *Hypsilobarbus curmuca* | 4 | EN |
| 26 | *Hypsilobarbus periyarensis* | 3 | EN |
| 27 | *Hypsilobarbus dubius* | 6 | EN |
| 28 | *Tor malabaricus* | 5 | EN |
| 29 | *Tor kudhree* | 9 | EN |
| 30 | *Osteochilus longidorsalis* | 2 | EN |
| | **Sub - Family: Danioninae** | | |
| 31 | *Salmophasia acinaces* | 1 | LC |
| 32 | *Barilius gatensis* | 16 | LC |
| 33 | *Barilius bakeri* | 10 | LC |
| 34 | *Barilius barana* | 2 | LC |
| 35 | *Barilius bendelisis* | 3 | LC |
| 36 | *Devario aequipinnatus* | 21 | LC |
| 37 | *Rasbora daniconius* | 13 | LC |
| | **Sub - Family: Oreininae** | | |
| 38 | *Lepiphygopsis typus* | 2 | EN |
| | **Sub - Family: Garrinae** | | |
| 39 | *Garra mullya* | 16 | LC |
| 40 | *Garra surendranathi* | 3 | EN |
| 41 | *Garra nastuta* | 1 | LC |
| 42 | *Garra periyarensis* | 2 | VU |
| 43 | *Garra hughi* | 3 | EN |
| 44 | *Garra gotyola stenorynchus* | 2 | LC |
| 45 | *Crossochelius latius latius* | 1 | LC |
| | **Family: Balitoridae** | | |
| | **Sub - Family: Balitorinae** | | |
| 46 | *Travancoria jonesi* | 8 | EN |
| | **Sub - Family: Nemacheilinae** | | |
| 47 | *Nemacheilus dennisoni* | 2 | LC |
| 48 | *Nemacheilus guntheri* | 7 | LC |



| | **Family: Cobitidae** | | |
|---|---|---|---|
| | **Sub - Family: Cobitinae** | | |
| 49 | *Lepidocephalus thermalis* | 5 | LC |
| | **Order: Siluriformes** | | |
| | **Family: Bagridae** | | |
| | **Sub - Family: Bagrinae** | | |
| 50 | ***Hemibagrus*** *punctatus* | 3 | CR |
| 51 | *Mystus cavasius* | 4 | LC |
| | **Family: Siluridae** | | |
| 52 | *Ompok bimaculatus* | 1 | NT |
| | **Family: Sisoridae** | | |
| | **Sub - Family: Glyptosterninae** | | |
| 53 | *Glyptothorax housei* | 1 | EN |
| | **Order: Cyprinodontiformes** | | |
| | **Family: Aplocheilidae** | | |
| | **Sub - Family: Aplocheilinae** | | |
| 54 | *Aplocheilus lineatus* | 3 | LC |
| | **Order: Synbranchiformes** | | |
| | **Sub- order: Mastacembeloidei** | | |
| | **Family: Mastacembelidae** | | |
| | **Sub - Family: Mastacembelinae** | | |
| 55 | *Macroganthus pancalus* | 1 | LC |
| 56 | *Mastacembelus armatus* | 1 | LC |
| | **Order: Perciformes** | | |
| | **Sub- order: Percoidei** | | |
| | **Family: Ambassidae** | | |
| 57 | *Chanda nama* | 2 | LC |
| | **Family: Pristolepididae** | | |
| 58 | *Peristolepis marignata* | 3 | LC |
| | **Sub- order: Labroidei** | | |



| | Family: Cichlidae | | |
|---|---|---|---|
| 59 | *Oreochromis mosambica* | 1 | NT |
| 60 | *Etroplus suratensis* | 3 | LC |
| 61 | *Etroplus maculatus* | 2 | LC |
| | **Sub- order: Gobioidei** | | |
| | **Family: Gobiindae** | | |
| | **Sub - Family: Gobiinae** | | |
| 62 | *Glossogobius guiris* | 1 | LC |
| | **Order: Mugiliformes** | | |
| | **Sub- order: Belonoidei** | | |
| | **Family: Belonidae** | | |
| 63 | *Xenetodon cancilia* | 3 | LC |
| | **Family: Hemiramphidae** | | |
| 64 | *Hyporhamphus limbatus* | 2 | LC |

* EX – Extinct; EW – Extinct in the Wild; CR – Critically Endangered; EN – Endangered; VU – Vulnerable; NT – Near Threatened; LRnt – Low Risk near threatened; LRlc – Low Risk least concern; LRcd – Low Risk conservation dependent; DD – Data Deficient.

**Table 3: Indices of diversity of fishes respective to altitudes of six river systems**

| Sampling Locations | Diversity (H') | Evenness (E) | Abundance | Richness (S) | Dominance (D) |
|---|---|---|---|---|---|
| Gulithuraipatti | 0.769 | 0.769 | 62 | 10 | 5.016 |
| Kallampalayam | 0.62 | 0.686 | 38 | 8 | 3.316 |
| Belemeenthurai | 0.841 | 0.932 | 19 | 8 | 8.55 |
| Orukomban range | 0.711 | 0.842 | 49 | 7 | 4.576 |
| Thenmudiparai | 0.74 | 0.875 | 59 | 7 | 4.833 |
| Baghapallam | 0.617 | 0.793 | 36 | 6 | 3.728 |
| Thellikal | 0.805 | 0.843 | 32 | 9 | 5.701 |
| Puliyarkutti 8th bridge | 0.879 | 0.921 | 39 | 9 | 7.8 |
| Puliyarkutti 3rd bridge | 0.401 | 0.841 | 17 | 3 | 2.429 |



| | | | | |
|---|---|---|---|---|
| Thunakadavu stream | 0.864 | 0.864 | 68 | 10 | 6.026 |
| Thunakadavu tunnel | 0.357 | 0.748 | 42 | 3 | 2.121 |
| Urilikal | 0.734 | 0.869 | 131 | 7 | 4.598 |
| Athirappalli | 1.01 | 0.936 | 52 | 12 | 11.143 |
| Pillapara | 0.718 | 0.923 | 25 | 6 | 5.769 |
| Kovaikutralam falls | 0.722 | 0.928 | 40 | 6 | 5 |
| Nellithurai | 0.757 | 0.896 | 29 | 7 | 5.639 |
| Oorpannikaham | 0.767 | 0.849 | 27 | 8 | 5.4 |
| Valukuparai | 0.91 | 0.954 | 28 | 9 | 9.947 |
| Melaparai | 0.798 | 0.944 | 19 | 7 | 7.773 |
| Naduthotam | 1.019 | 0.915 | 152 | 13 | 9.936 |
| Tmmikuppamthodu | 0.527 | 0.678 | 41 | 6 | 2.384 |
| Sorrakottaodai | 0.465 | 0.976 | 16 | 3 | 3.243 |
| Mullaithodu | 1.045 | 0.968 | 48 | 12 | 12.966 |
| Anjurily | 0.537 | 0.768 | 19 | 5 | 3.054 |
| Thenkasithodu | 0.638 | 0.668 | 100 | 9 | 3.327 |
| Kadapilliyarthittu | 0.8 | 0.886 | 37 | 8 | 6.055 |
| Belikoondu | 0.625 | 0.804 | 16 | 6 | 3.75 |
| Nadathittu | 1.198 | 0.921 | 77 | 20 | 15.481 |
| Sinnaru | 1.268 | 0.959 | 75 | 21 | 21.346 |
| Thonanthikla | 1.069 | 0.909 | 46 | 15 | 11.129 |
| Noolpuzha | 0.946 | 0.946 | 78 | 10 | 8.938 |



**Table 4: Species composition among the 31 sites**

| Cluster no | Cluster between | Study sites |
|---|---|---|
| 1 | 1 - 4 | Thunakadavu stream, Baghapallam, Kallampalayam, Thunakadavu tunnel |
| 2 | 5 -7 | Thenmudiparai, Orukomban range, Gulithuraipatti |
| 3 | 8 - 28 | Melaparai, Valukuparai, Belemeenthurai, Anjurily, Oorpannikaham, Nellithurai, Belikoondu, Kadapilliyarthittu, Sorrakottaodai, Puliyarkutti 3rd bridge, Mullaithodu, Kovaikutralam falls, Puliyarkutti 8th bridge, Sinnaru, Nadathittu, Thonanthikla, Thellikal, Pillapara, Athirapalli, Noolpuzha, Ummikuppamthodu |
| 4 | 29 | Naduthotam |
| 5 | 30 | Thenkasithodu |
| 6 | 31 | Urilikal |

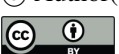



**Table 5: Distribution and abundance of fishes of six river systems**

Collection sites (Collection site number as in Table 1)

| S.no | 1 | 2 | 3 | 4 | 5 | 6 | 7 | 8 | 9 | 10 | 11 | 12 | 13 | 14 | 15 | 16 | 17 | 18 | 19 | 20 | 21 | 22 | 23 | 24 | 25 | 26 | 27 | 28 | 29 | 30 | 31 |
|---|---|---|---|---|---|---|---|---|---|---|---|---|---|---|---|---|---|---|---|---|---|---|---|---|---|---|---|---|---|---|---|
| 1 | 1 | | | 4 | 6 | 12 | 7 | 10 | 5 | 11 | | 32 | 5 | 2 | 5 | | | 2 | 2 | | | 5 | 4 | 5 | 2 | 4 | | 9 | 5 | 2 | 10 |
| 2 | | 1 | 5 | | | | 1 | | 2 | 6 | | | | | | | | | 2 | | | | | | | | | | 5 | 2 | |
| 3 | | 2 | | | | | 1 | | | | | | | | | | | | | | | | | | | | | | 2 | | |
| 4 | | | | | | | | 5 | 10 | 5 | | 13 | | | 15 | | 1 | 1 | | 2 | | 4 | 5 | | | | | | 2 | | 15 |
| 5 | 1 | | | | | | | | | | | | | | | | | | | | | | | | | | | 5 | 2 | 5 | |
| 6 | 15 | 1 | | | | | | | | | | | | | | 10 | | | | | | | | | | | | | 6 | | |
| 7 | 1 | 1 | | | | | | | | | | | | | | | | | | | | | | | | | | | | | |
| 8 | 7 | 15 | | | | | | | | | | | | | | | | | | | | | | | | | | | | | |
| 9 | | | | | | | | | | | | 11 | | | | | | | | | | | | | | | | | | | |
| 10 | | | | | | | | | | | | | | | | | | | | 19 | 26 | | | | | | | | | | |
| 11 | | | | | | | | | | | 25 | | 8 | 5 | 5 | 5 | | | | | | | | | | 4 | | 3 | 3 | 5 | |
| 12 | | | | | | | | | | | | | | | | 10 | | | | | | | | | | 10 | 1 | 3 | | | |
| 13 | | | | | | | | | | | | | 5 | 5 | | | | | | | | | | | | | | | | | |
| 14 | | | | | | | | | | | | | 1 | 1 | | | | | | | | | | | | | | | | | |
| 15 | | | | | | | | | | | | | | | | 3 | | | | | | | | | | | | | | | |
| 16 | | | | | | | | | | | | 10 | | | | | | | | | | | | | 2 | | | | | | 10 |
| 17 | | 1 | | | | | | | | | | | | | | | | | | | | | | | | | | | | | |
| 18 | | | | | | | | | | | | | | | | | 1 | | | | | | | | | | | | 3 | | |
| 19 | | | | | | | | | | | | | | | | 2 | | | | | | | | | | | | | | | |
| 20 | | | | | | | | | | | | | | | | | | | | | | | | | | | | | | | |
| 21 | | | | | | | | | | | | | | | | | | | | | | | | | | | | | | | |
| 22 | | | | | | | | | | | | | | | | 1 | | | | | | | | | | 1 | | 1 | | | |
| 23 | | | | | | | | | | | | | | | | 1 | | | | | | | | | | | 1 | | | | |
| 24 | | | 2 | | | | | | | | | | | | | | | | | | | | | | | | | | | 1 | |
| 25 | | | | | | | | | | | | | | | | | | | | 10 | | | 7 | 1 | | | | | | | 2 |
| 26 | | | | | | | | | | | | | | | | | | 2 | | 15 | | | 3 | | | | | | | | |
| 27 | | | | | | | | | | | | | | | | | | | 1 | 17 | | | | | | | 1 | 3 | 3 | | |
| 28 | | | | | | | | | | | | | | | | | | | | 10 | | | 2 | | | | | 2 | 4 | | |
| 29 | | | | | 2 | | 2 | | | | | | | | | | 5 | 3 | 2 | 5 | | | | | | | | 2 | 3 | 3 | |
| 30 | | | | | | | | | | | | | | | | | | | | 5 | | | | | | | | | | 4 | |
| 31 | | | | | | | | | | | | | | | | | | | | | | | | | | | | 4 | | | 4 |



| Row | C1 | C2 | C3 | C4 | C5 | C6 | C7 | C8 | C9 | C10 | C11 | C12 | C13 | C14 | C15 | C16 | C17 | C18 | C19 | C20 | C21 | C22 | C23 | C24 | C25 | C26 | C27 | C28 | C29 | C30 |
|---|---|---|---|---|---|---|---|---|---|---|---|---|---|---|---|---|---|---|---|---|---|---|---|---|---|---|---|---|---|---|
| 32 | 10 |  |  |  |  |  | 20 |  | 5 |  | 4 | 20 |  | 2 |  | 3 | 4 |  | 5 |  |  | 5 | 5 | 3 |  | 11 | 18 | 4 |  | 4 |
| 33 | 8 |  |  |  |  |  | 21 |  |  |  |  | 15 |  |  |  | 5 | 5 |  | 2 |  |  |  | 2 |  |  | 8 | 2 |  |  |  |
| 34 |  |  |  |  |  |  |  |  |  |  |  |  |  |  |  |  |  |  |  |  |  | 5 | 2 |  |  |  |  |  |  |  |
| 35 |  | 3 | 3 | 3 |  | 2 |  |  |  |  |  |  |  |  |  |  |  |  |  |  |  |  |  |  |  |  |  |  |  |  |
| 36 |  | 5 | 2 | 10 |  |  |  | 10 |  |  | 2 | 10 | 2 |  |  |  | 6 | 5 | 5 | 47 | 15 | 23 | 6 | 4 | 14 | 7 | 5 | 2 | 15 | 11 |
| 37 |  |  | 7 | 2 |  |  | 1 |  |  |  | 5 | 2 |  | 4 |  |  |  |  | 4 | 14 |  |  |  | 1 |  | 3 | 7 |  |  |  |
| 38 |  | 11 | 4 | 3 |  | 2 |  | 2 | 2 |  |  | 25 |  |  | 10 |  |  | 7 |  |  |  | 7 | 3 | 11 | 4 | 22 | 12 |  | 2 | 20 |
| 39 |  |  |  |  |  |  | 3 |  | 6 |  |  |  |  |  |  |  |  |  | 8 |  |  |  |  |  |  |  |  |  |  |  |
| 40 |  |  |  |  |  |  |  |  |  |  |  |  |  |  |  |  |  |  | 6 |  |  |  |  |  |  |  |  |  |  |  |
| 41 |  |  |  |  |  |  |  |  | 2 |  |  |  |  |  | 3 |  |  |  |  |  |  |  |  |  |  |  |  |  |  |  |
| 42 |  |  |  |  |  |  |  |  |  |  |  |  |  | 4 |  |  |  |  |  |  |  |  |  |  |  |  |  |  |  |  |
| 43 |  |  |  |  |  |  |  |  |  |  | 2 |  | 5 | 5 |  |  |  |  |  |  |  |  |  |  |  |  |  |  |  |  |
| 44 |  |  |  |  |  |  |  |  |  |  |  |  |  |  |  |  |  |  |  |  |  |  |  |  |  |  |  |  |  |  |
| 45 |  |  |  | 4 | 2 |  |  |  |  |  |  |  |  |  | 1 |  |  |  |  |  |  |  |  |  | 1 |  |  | 1 |  |  |
| 46 |  |  |  |  |  |  | 2 |  |  |  | 2 |  | 4 | 5 | 4 |  | 5 |  |  |  |  |  | 5 |  |  |  |  |  |  |  |
| 47 |  |  |  |  |  |  |  |  |  |  |  |  | 3 |  |  |  |  |  |  |  |  |  |  |  |  |  |  |  |  |  |
| 48 |  |  |  |  |  |  | 2 |  | 4 |  |  |  |  |  |  |  |  |  |  |  |  |  |  |  | 4 |  |  |  |  |  |
| 49 |  |  |  |  |  |  |  |  | 3 |  |  | 2 |  |  |  |  |  |  | 2 |  |  | 1 | 1 | 2 |  |  |  |  |  |  |
| 50 |  |  |  |  |  |  |  |  |  |  |  |  |  |  |  |  |  |  |  |  |  |  |  |  |  |  | 1 | 2 |  |  |
| 51 |  |  |  |  |  |  |  |  |  |  |  |  |  |  |  |  |  |  | 1 |  |  |  |  |  |  |  |  | 2 |  |  |
| 52 |  |  | 3 | 7 |  |  |  |  |  |  |  |  |  |  |  |  |  |  |  |  |  |  |  |  | 1 |  |  |  |  |  |
| 53 |  |  | 1 | 1 |  |  |  |  | 5 | 7 |  |  |  |  |  |  |  |  |  |  |  |  |  |  |  |  |  |  |  |  |
| 54 | 4 | 2 |  |  |  |  |  |  |  |  |  |  |  |  |  |  |  |  |  | 4 |  |  |  |  |  |  |  |  |  |  |
| 55 |  |  |  |  |  |  |  |  |  |  |  |  |  |  |  |  |  |  |  |  |  |  |  |  |  |  |  |  |  |  |
| 56 |  |  | 2 |  |  |  |  | 1 |  |  |  |  |  |  |  |  |  |  |  |  | 2 | 3 |  |  |  |  |  |  |  |  |
| 57 | 10 | 1 |  |  |  |  |  |  |  |  |  |  |  |  |  |  |  |  |  |  |  | 2 |  |  |  |  |  |  |  |  |
| 58 |  |  | 7 |  |  |  |  |  |  |  |  |  |  |  |  |  |  |  |  |  |  |  |  |  |  |  |  |  |  |  |
| 59 |  |  |  | 1 | 8 | 10 |  |  |  |  |  |  |  |  |  |  |  |  |  |  |  |  |  |  |  |  |  |  |  | 1 |
| 60 |  |  |  | 10 |  | 4 |  |  |  |  |  |  |  |  |  |  |  |  |  |  |  |  |  |  |  |  |  |  |  |  |
| 61 |  |  |  |  |  |  |  |  |  |  |  |  |  |  |  |  |  |  |  |  |  |  |  |  |  |  |  |  |  |  |
| 62 |  |  | 5 |  |  |  |  |  |  |  |  |  |  |  |  |  |  |  |  |  |  |  |  |  |  |  |  |  |  |  |
| 63 |  | 2 | 2 | 2 |  |  |  |  |  |  |  |  |  |  |  |  |  |  |  |  |  |  |  |  |  |  |  |  |  |  |
| 64 |  | 1 |  | 2 |  |  |  |  |  |  |  |  |  |  |  |  |  |  |  |  |  |  |  |  |  |  |  |  |  |  |



**Table 6: Water quality of 31 study sites of six river systems**

| Index | pH | Conductivity (mS) | TDS (ppm) | Resistivity (KΩ) | DO (mg/L) | Salinity (ppt) | Water temperature (°C) |
|---|---|---|---|---|---|---|---|
| Gulithuraipatti | 8.4 | 57.8 | 20.37 | 24.2 | 3.5 | 0.03 | 23.8 |
| Kallampalayam | 7.9 | 45.2 | 28.5 | 21.9 | 2.5 | 0.02 | 24.1 |
| Belemeenthurai | 8.4 | 59.2 | 37.7 | 16.4 | 1.3 | 0.03 | 24.5 |
| Orukomban range | 7.5 | 33.9 | 26.5 | 22.4 | 3.5 | 0.02 | 23.4 |
| Thenmudiparai | 8 | 45.2 | 28.5 | 21.9 | 2.5 | 0.02 | 24.1 |
| Baghapallam | 8 | 57.8 | 38.0 | 16.8 | 2.4 | 0.03 | 21.7 |
| Thellikal | 8.8 | 59.2 | 37.7 | 16.4 | 1.3 | 0.03 | 24.5 |
| Puliyarkutti 8[th] bridge | 7.79 | 27.8 | 18.0 | 34.8 | 5.4 | 0.01 | 23.5 |
| Puliyarkutti 3[rd] bridge | 7.79 | 27.8 | 18.0 | 34.8 | 5.4 | 0.01 | 23.5 |
| Thunakadavu stream | 5.9 | 38.3 | 28.3 | 22.2 | 5.09 | 0.02 | 21.4 |
| Thunakadavu tunnel | 5.9 | 38.3 | 28.3 | 22.2 | 5.09 | 0.02 | 21.4 |
| Urilikal | 7.2 | 78.7 | 51.9 | 12.9 | 1.4 | 0.03 | 24.1 |
| Athirappalli | 7.2 | 35.2 | 47.5 | 3.97 | 0.73 | 0.02 | 32.7 |
| Pillapara | 7.6 | 34.0 | 19.5 | 29.9 | 0.89 | 0.02 | 33.6 |
| Kovaikutralam falls | 7.5 | 31.3 | 20.1 | 32.3 | 3.2 | 0.01 | 22.5 |
| Nellithurai | 7.3 | 30.3 | 20.3 | 31.5 | 2.3 | 0.01 | 25.5 |
| Oorpannikaham | 8.3 | 50.3 | 32.3 | 20.0 | 1.2 | 0.02 | 24.8 |
| Valukuparai | 7.7 | 66.9 | 43.8 | 15.1 | 0.7 | 0.03 | 24.8 |
| Melaparai | 9 | 44.7 | 28.8 | 22.5 | 1.3 | 0.02 | 26.1 |
| Naduthotam | 7.5 | 46.2 | 30.4 | 20.6 | 0.7 | 0.01 | 25.9 |
| Ummikuppamthodu | 7.7 | 64.9 | 43.2 | 17.1 | 1.2 | 0.03 | 24.8 |
| Sorrakottaodai | 8 | 34.2 | 21.9 | 29.5 | 1.1 | 0.01 | 23.1 |
| Mullaithodu | 8.1 | 78.6 | 51.4 | 12.5 | 0.9 | 0.04 | 24.2 |
| Anjurily | 7.2 | 21.5 | 13.6 | 47.5 | 4.86 | 0.01 | 19.2 |
| Thenkasithodu | 5.2 | 22.0 | 13.7 | 45.6 | 6.11 | 0.01 | 18.9 |
| Kadapilliyarthittu | 9.6 | 39.1 | 26.3 | 2.58 | 0.72 | 0.18 | 30.5 |
| Belikoondu | 9.4 | 39.8 | 26.3 | 2.63 | 0.63 | 0.17 | 32.7 |
| Nadathittu | 9.4 | 39.8 | 26.3 | 2.63 | 0.63 | 0.17 | 32.7 |
| Sinnaru | 9.2 | 39.5 | 26.3 | 2.65 | 3.11 | 0.11 | 30.2 |
| Thonanthikla | 9.2 | 39.5 | 26.3 | 2.65 | 3.11 | 0.11 | 30.2 |
| Noolpuzha | 7.32 | 85.2 | 51.7 | 11.8 | 3.62 | 0.04 | 23.2 |





Fig 1: Collection location of six river systems



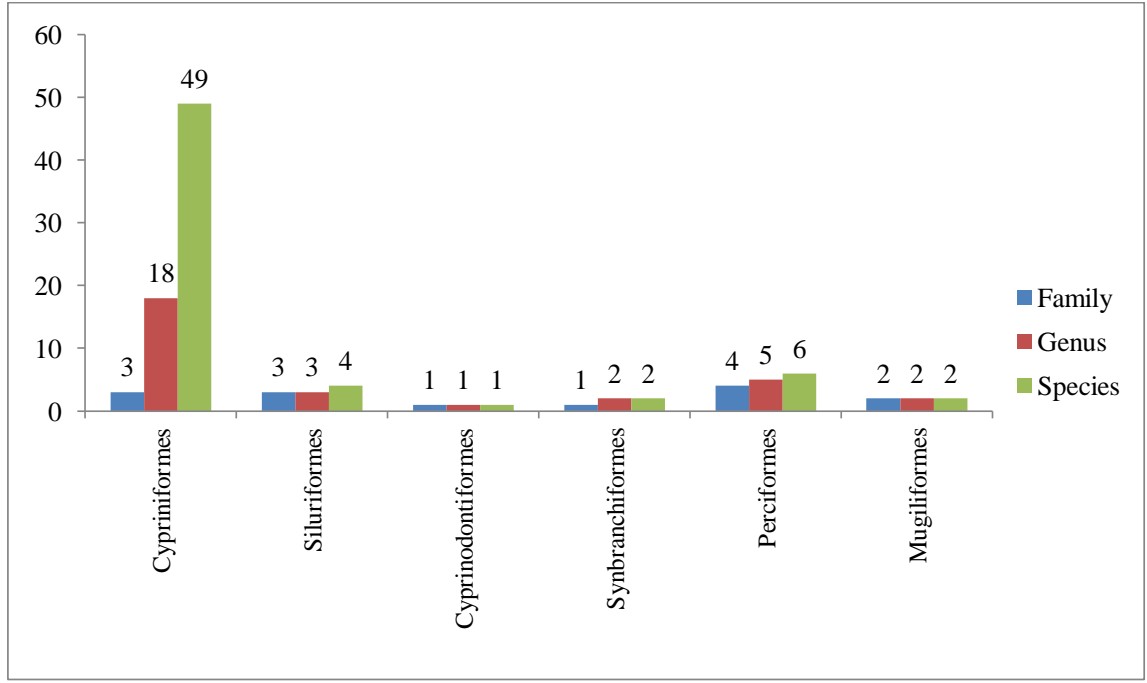

**Fig. 2. Representation of fishes in different order**

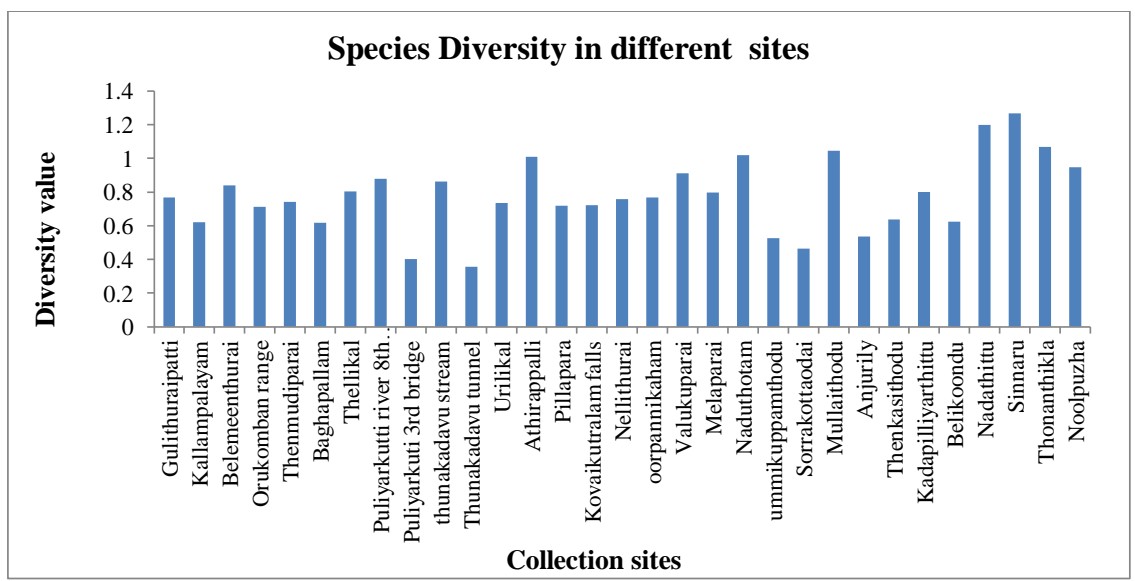

**Fig 3: Species diversity in among 31 sites**

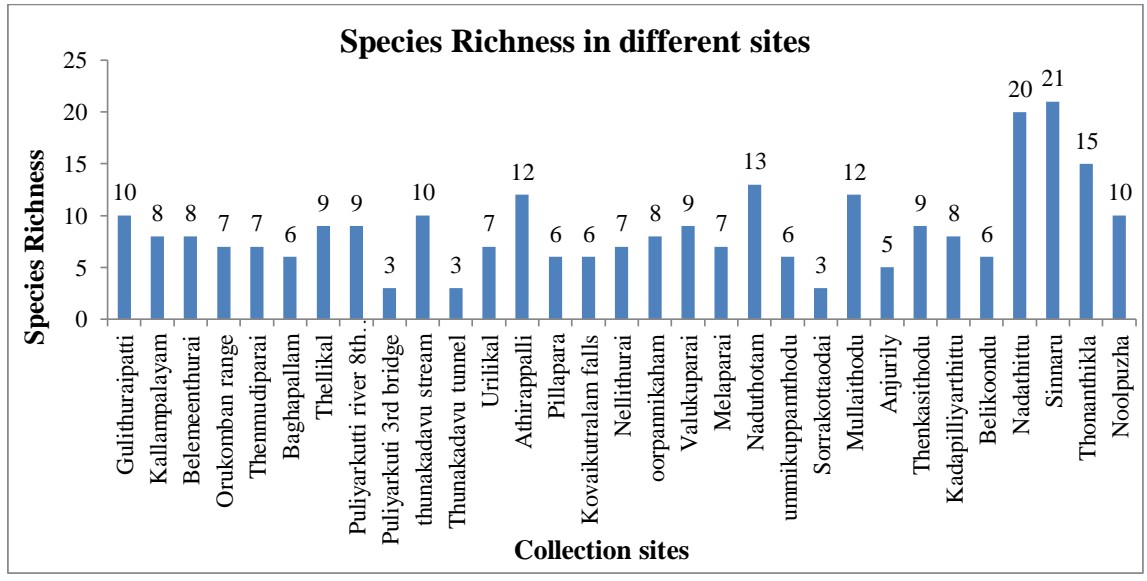

Fig 4: Species richness among 31 sites

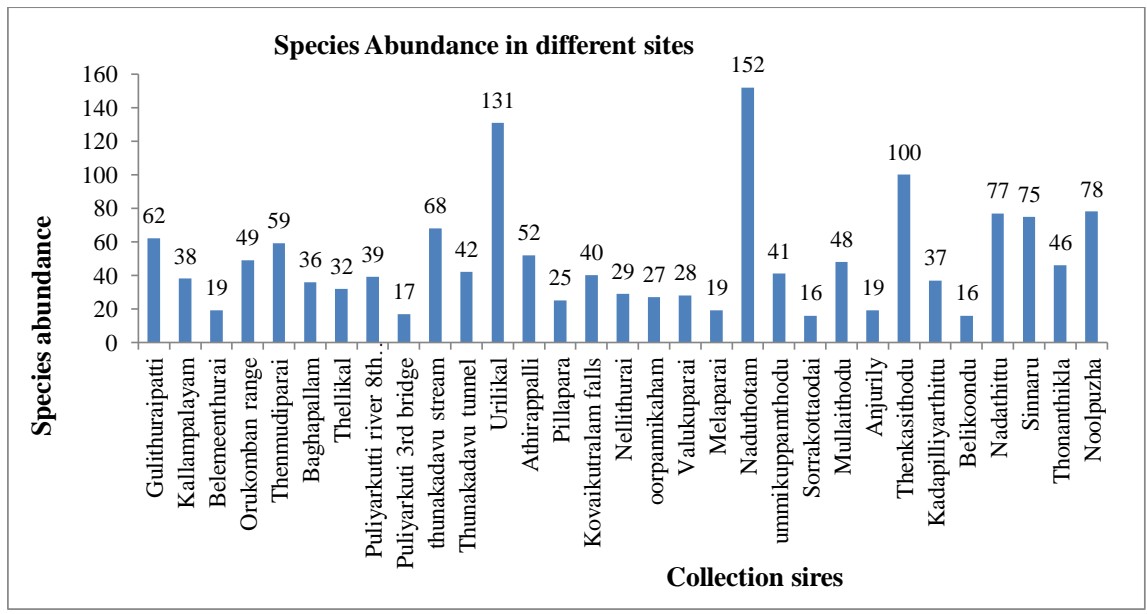

Fig 5: Species abundance in among 31 sites

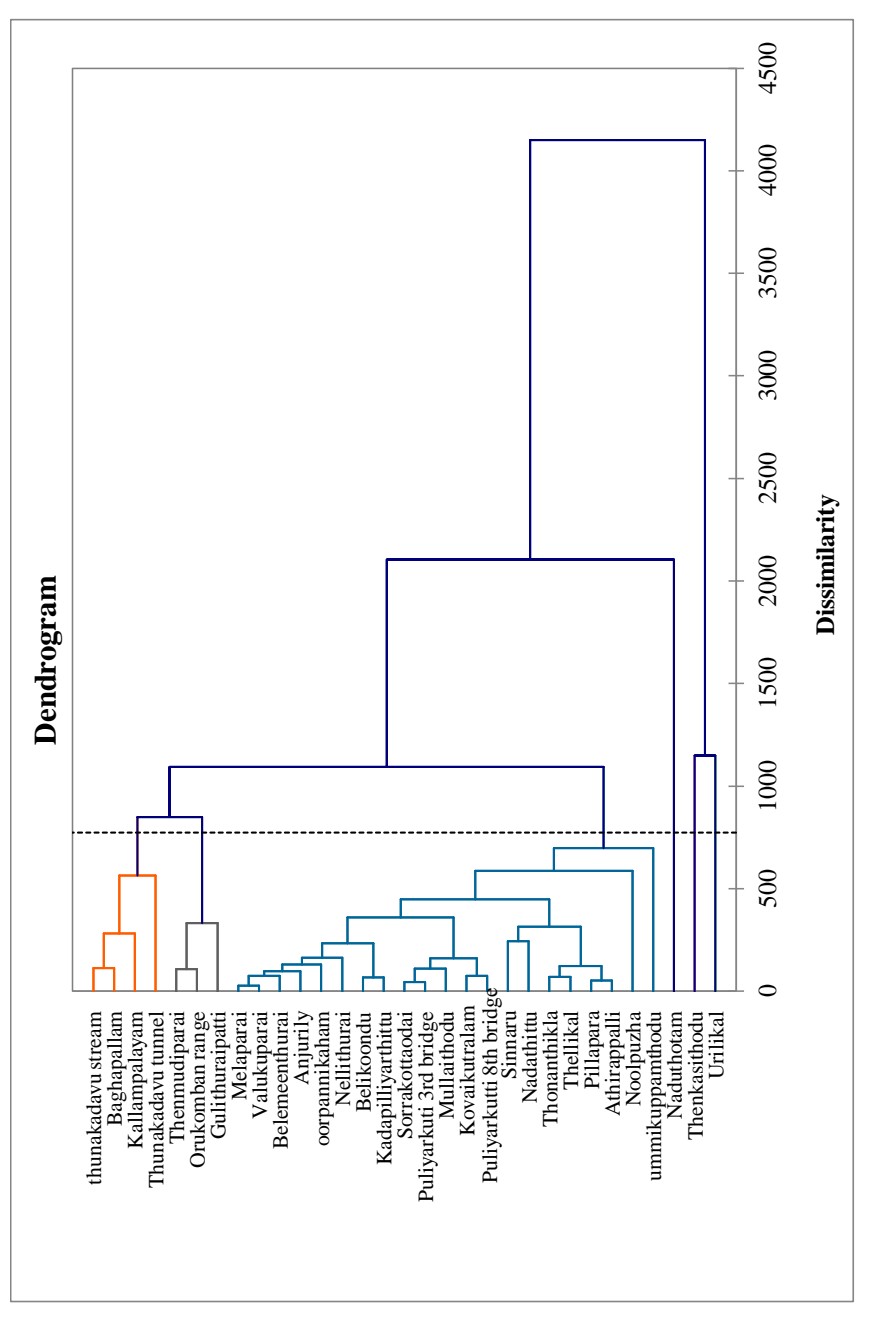

**Fig 6: Cluster dendrogram shows the dissimilarity between 31 sites**



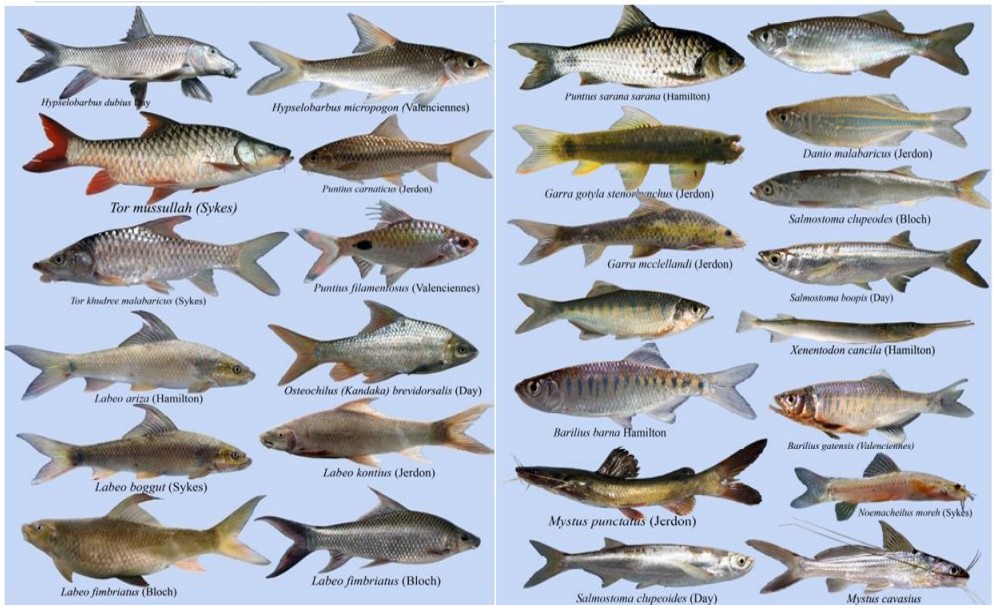

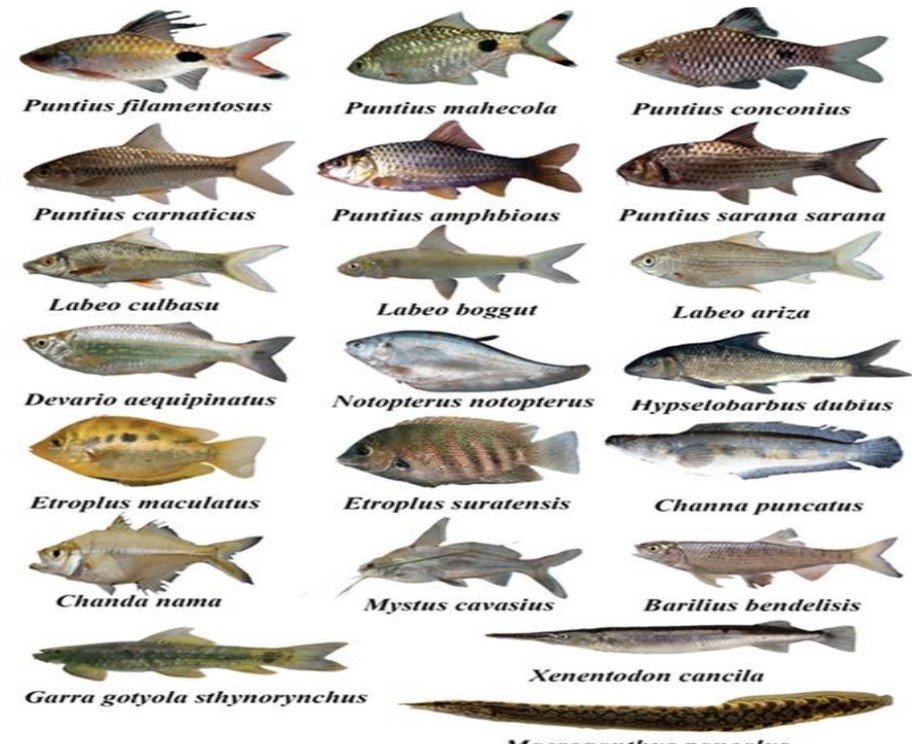

**Fig. 7: Fishes collected from various water bodies of SWG**