# Peer review of "FRESHWATER FISH FAUNA OF RIVERS OF SOUTHERN WESTERN GHATS, INDIA"

_Earth System Science Data, 2017_

## Editor Comment (EC1) · F. Huettmann (Editor) · 1 Jan 2018

Dear Colleagues,

I find this manuscript (MS) convinces in the fact that it provides a species overview and does it all as Open Access. It's short and concise.

I value those efforts highly, and based on our own experience in that wider region.

To me, this MS is to be accepted after considering some comments below.

I find, this style of research - open access- breaks new ground and is a great contribution in many regards. The analysis done looks robust to me, as presented (please

[Figure]

mention more software details, if you can).

The authors state already a lack of data and few surveys for the area, and so this work matters even more then. The Ghats are a global biodiversity hotspot and deserve more modern online research for sure!

This work also helps addressing the Satpura hypothesis, which is part of modern biogeography and for India overall.

While water (and sand in rivers) is the 'new oil', the authors can easily make that link to environmental impact studies more; they should.

This work further links with BarCoding and DNA work (but not with Genbank). Ideally, the voucher specimen should link with a URL where to find them online and how to connect them further. This would allow for a more global access, too.

Due to the mesh-size issues of the compiled data, this data set should be treated probably as a first minimum estimate (similar to 'presence only' data, I would say). I would simply interpret it that way. Documenting the research design is very relevant for a better interpretation of those data.

So far, I see the data not much presented as a data base (just tables). So I would encourage the authors to serve a digital database copy of all data (ideally with metadata).

The geo-referencing in the table is very important and great, but seems to lack the 'seconds'. Thus, it's still great but not very precise (=>the minutes need decimals or seconds details). This should be stated and what it means, on the ground for an accuracy for a user of this data. To be professional, each location needs an error estimate (see GBIF.org data for an example, or biogeomancer tool and PLOS1 paper https://www.gbif.org/document/80536/biogeomancer-guide-to-best-practices-in-georeferencing ).

In addition, data collections should ideally be time/date-referenced as well; if possible. I would elaborate on that either way (I assume it's not mentioned because data are

compiled; say so then).

As a great addition, I like the physical data collection and water descriptions.

I find table 5 to be clearly a highlight of this study and data set!

The fish taxonomy issues are well mentioned. What is a little bit missing though is a good and clear taxonomy citation that is followed here. Ideally, it probably should be ITIS.gov (which can take species list uploads and which provides taxonomic serial numbers TSNs) but for now, any other will do fine if stated clearly. It's a nice start for Open Access work.

I assume the first english species descriptions do not match so well with recent species IDs, nor are they available online. This can be mentioned; same applies to the evolutionary questions in fish taxonomy (which appear to be quite fundamental ?).

Smaller additional comments for improvements:

The abstract should present details mentioned in the last phrases.

Line 74: Should read Kolkata

Line 192: Should have a lower case name

Line 209-210: This should need more details.

Fig 7 Should state source/author of the drawings.

Tables should have the columns correctly formatted.

This concludes my first assessment.

Once more, I really like this MS and the work and find it can easily make for a nice contribution indeed, now as a modern Open Access online manuscript for a global audience with suitable impact. Thanks.

Kind regards Falk Huettmann PhD, Associate Professor University of Alaska Fairbanks

(UAF), USA

---

## Author Comment (AC1) · 8 Jan 2018

Title: Freshwater Fish Fauna of Rivers of Southern Western Ghats, India Manuscript ID : essd-2017-118

Dear Prof. Falk Huettmann:

Thank you very much for your kind consideration and help to our manuscript! According to your suggestions, we added all suggestions given by you for improvement. Meanwhile, the formats of all tables and figures were revised with the journal's requirements.

Thanks again for the reviewers and the editor for your kind consideration and help!

[Figure]

Expecting my manuscript for acceptance in the ESSD journal.

Best regards

Sincerely yours,

Anbu Aravazhi Arunkumar and Arunachalam Manimekalan

Please also note the supplement to this comment:
https://www.earth-syst-sci-data-discuss.net/essd-2017-118/essd-2017-118-AC1-supplement.pdf

––––––––––––––––––––––––––

**Supplement:**

[revised manuscript text omitted]

genus *Puntius* is splitted into four genera like *Systomus, Dawkinsia, Dravidia and Pethia* which makes the genus
still under inconclusive status. In the present study, the fish species were collected by using different mesh size
of gill nets, cast net and dip nets from the long and meandering eastward flowing river systems of Bhavani, Moyar,
Chalakudy, Periyar, Cauvery and Kabini. Species from Southern Western Ghats have a confusing taxonomy and
exhibit a clear morphological variation within and between the species. The species collected in different
geographical locations did not express variations in body patterns or in colorations. The species like *Puntius*
*fasciatus and Puntius melanampyx* seems to be the same species temptationally but it turns out to be separate in

[revised manuscript text omitted]

*© Dr. A. Manimekalan & Dr. A. A. Arunkumar*

**Fig. 7: Fishes collected from various water bodies of SWG**

---

## Referee Comment (RC1) · Anonymous Referee #1 · 26 Jan 2018

Review ESSD-2017-118

Somewhat surprised to see this paper submitted to ESSD. But a good direction for the journal and compliments to the authors for attempting to make their data publicly accessible. I share the authors view that they address a critical region of our planet.

Fundamentally I regard the data gathering, species identification and data analysis as sound. As this review will show I have only a few comments and suggestions on those aspects. I believe however that the data and the paper need a stronger and clearer presentation to appeal to a wide audience. On those presentation aspects I have made many comments and suggestions, enough perhaps to require a substantial revision. If the editors agree I will suggest a major revision that should lead to eventual publication.

The Pangaea link works very well. The authors basically repeat the tables from the manuscript as data files in the Pangaea archive. Because the Pangaea landing page correctly identifies the four data tables (tables 1, 3, 5 and 6) as tab-delimited files, the names of the files in the dataset folder should carry this designation. E.g. Arunkumar-etal_2015-T3.tsv rather than Arunkumar-etal_2015-T3.tab. R, Excel and other spreadsheets can easily ingest a .tsv file but those software packages will not recognise a .tab file. Once I renamed them I had no problem to open all the .tsv files.

Lines 13, 14. This sentence at the start of the abstract, about studying the freshwater fish from 2010 to 2013 can give a wrong impression. Some readers will assume that you studied these rivers in repetitive years, e.g. in 2010, again in 2011, out to 2013. In fact this study reports the outcome of a collection and identification process that covered 31 separate sampling sites that required 4 years (2010 to 2013) to complete. Nowhere in the documentation do we read about any repeat sampling. You should make very clear that you conducted one comprehensive sampling and assessment of each site in a process that required 4 years, 2010 to 2013.

Lines 16, 17. These lines, about 64 species, some many orders, families, etc. repeat information from line 14. Remove the sentence in line 14? We do not need to see this information twice, so close together.

Line 18. Plural of 'genus' should appear as 'genera' (as in line 17 above)?

Lines 23 to 26. The collection and identification of some fish species considered endangered or critically endangered probably represents an import contribution of this study. These fragmented and confused statements do not provide an adequate summary. The manuscript that follows probably needs an explicit section on endangered species and the significance of finding them in the Western Ghats. This abstract implies a "concise discussion" but that discussion never appears in the subsequent manuscript?

The introductory paragraphs look good.

Line 45, 46. The "Satpura" hypothesis? Specific to Indian or South Asian ecosystems or something more general. How and why did that hypothesis stimulate discussion on, for example, "endemism". Does this discussion have relevance to this paper?

Lines 48 to 62, This paragraph about new species discoveries has relevance to the larger point about the Western Ghats as a location and source of unique biodiversity? We need to know the reason for this list of species names.

Line 65, 66. This combination of various types of nets has some minimum capture size? Larger organisms could avoid the nets? Please give users a sense of the size range captured by your sampling. What happens to any invertebrates? Discarded, recorded, ignored, or not captured?

Line 67. Five specimens from each species. This means you did some species identification in the field? Did you have a statistical basis for this sampling strategy? Or a valid logistical constraint? By focusing on repeatable numbers of species present, you have minimised the presence-absence question? The issue of absence of expected species does not arise in this manuscript, despite earlier mention of disturbance and invasives? Perhaps this discussion

belongs in a subsequent research paper but those future researchers will need to know how to understand your data.  Absence means zero specimens of a given species collected or fewer than 5 specimens collected?

Line 81, water samples collected "post-monsoon".  We should have actual dates for all collection episodes, fish and water?  Perhaps these exist in your database but they do not appear in any of the data shared here.

Lines 98, 99.  Confusion here for the reader about primary sources of uncertainty.  One source, identified here, involves conversion of information from "elementary" or "original" data sheets?  Which feeds which?  Which represents inputs to the database?  Errors frequent or rare in these data translation processes?  We need more clarification here.  Later, in the summary, you should list for users all the known sources of uncertainty.

Line 104.  Present collection sites based on prior literature reports?  Shouldn't we have this information, along with discussion of impact on reliability and repeatability, earlier, in the methods section?  If true, this opens the possibility of comparison of abundance, species presence-absence, etc., with earlier collections?  We need to know if, where, and to what degree the present collections enable these historical comparisons.  We do not need the comparisons themselves - those perhaps belong in a separate research paper - but we do need to know if the current collections enable such comparisons with prior collections.  If not, why not?

Lines 104 to 112. This description of geomorphology and biodiversity of the Western Ghats belongs in the introduction?

Line 114.  We need a much better presentation of the sampling sites.  Figure 1 shows only 16 out of 31 sites.  If the authors want to show locations on a map, we need a better, more complete map.  In Table 1 we get very useful information of sites by river system with elevation and lat lon for each site. But in Table 5 and 6 we lose that information.  Those tables should still make clear the association of sites with rivers; you do not want us as users making mistakes in our assignment of site to river system.  Table 1 does not seem to sort the site by elevation; somewhere we need that information.  In the .tsv data file for table 1 I could sort by river system and then elevation (or forest type); better the authors should do this for all readers?

Line 119.  Discussion of diversity, abundance and distribution by site, but not by river system, elevation, drainage area, proximity to human influence, presence or absence of hydroelectric dams, etc.  Later in this results section the authors mention elevation, water temperature, water quality, lakes, etc.  But here we don't get any sense of those factors for all sites or for each river system; we only see site-by-site lists.

Line 128.  Species similarity was very 'low' rather than very "less"?  The authors only hint at all the factors that might impact similarity, e.g. as plotted in Figure 6.  Again we would need to see those sites according to their river systems rather than independently?  Or, do the authors imply that site-to-site differences exceed river-to-river differences?  We do not get a clear treatment of river vs site data and differences that would allow us to assess the validity of such similarities or differences.  A clearer presentation of sites by river system, and of summary statistics of river vs river, or of the absence of significant differences, would help!

Line 138. IS: 10500 Permissible limits.  This represents an India-wide water quality standard?  We need a reference to it?

Lines 137 to 154.  This water quality discussion occurs almost entirely by site, not by river system.  Why?  Here and in Table 6, I feel surprised to see salinity.  Conductivity I expect, and perhaps resistivity, but what do these very low values of 'salinity' represent and report?  Do we have a definition of the salts involved?  Not the typical seawater salts, presumably, so we must have a different freshwater definition of salinity?  In other ESSD papers reporting mountain stream water quality, they typically do not report salinity.  Because all these values lie below 0.5 ppt (one commonly-accepted definition of freshwater) we should consider them all to have negligible amounts of salts?  If the authors do not make explicit use of the salinity data, we do not need them?

Lines 156 to 210.  After the water quality paragraph the authors provide four paragraphs on species appearance, diversity, abundance, etc. without clear conclusion and particularly without confirmation of Western Ghats as a biodiversity hotspot or as a 'refuge' for endangered species. Either we need less of species lists or we need more synthesis and assessment of the fundamental question: will these data allow and support discussion and conclusion about regional biodiversity and biological refuges or not - even if those discussions occur in other research papers or in other conservation fora.  Readers need to get from the authors a sense of confidence on how to use these data!  Unfortunately, from this somewhat confused and random discussion of physical and biological influences on habitats and species presence or abundance, we fail to get a clear understanding of the authors' confidence in their own data.  We also learn that the Periyar river flows westward when earlier we read that this study focused on eastward-flowing rivers and about the existence of Periyar Lake.  Certainly the map in Figure 1 and none of the text so far gave us any hints of lakes.

Line  188: "moolavaigae"?  Presumably this refers to a high-elevation location of Moolavaigae?

Line 212, 213, Summary.  I do not understand the point of the first sentence about morphological variation related to micro- or macro-habitat?

Lines 220 to 222.  Again by individual sites, with no reference to river systems.

Line 223: The present study failed to convince this reader that altitude had any consistent effect. Probably more related to a weakness in the presentation rather than a weakness in the data, but in either case the paper has not shown us convincing data relating biodiversity to elevation.

Lines 224 to 232: Perhaps valid statements toward a positive "ecological spirit", but in fact from these data the authors have not guided us to conclusions about "sharp decline" (line 227) or about social pressures.

A good conclusion should briefly summarise the data, explicitly caution users about uncertainties and limitations of the data, and then outline both the present impacts (for conservation management) and need or intention for future monitoring or data gathering.

Table 1.  'Forest Type': these terms come from FAO or GBIF definitions or from an India Forest Classification scheme?  Readers need to know how to relate this terminology to other data from other regions.  'Stream Order': hydrologists will understand this general mechanism to indicate stream branching but the authors should specify whether these represent standard Strahler stream order numbers or some other India-specific index of stream branching?  'Area' presumably represent catchment area above (upstream) of the sampling location.  The authors should inform readers how they calculated this or from what source they extracted this information.  Likewise for 'Volume', this presumably represents annual mean volume measured at some exit or drainage point of each stream and river?  Again, readers need to know where this information comes from.  In this table, 'Mean Velocity' represents a code referenced in a footnote, and not an absolute value?  The authors could reduce confusion by using the actual codes 'slow', 'moderate', 'very fast', etc?

Table 2.  Comprehensive species list but the IUCN codes in column 4 do NOT match the descriptions in the footnote.  For example, the footnote does not define LC as it appears in the Table while 'LRnt' from the footnote never appears in the Table.

Table 3.  We need these data explicitly organised by river system and perhaps sorted in order of elevation within each river system.  With effort a user can establish these distinctions and filter by elevation by using the .tsv file but the authors need these improvements in Table 3 in order to support their discussion about, for example, biodiversity and altitude?

Table 5.  Species by generic number code vs. site by similar generic number code.  Inclusion of actual species names (as genus.sp) and clearer organisation by river system would greatly increase the utility and information content of this Table!

Table 6.  Again, organise this by river system and sort by elevation?

Figure 1.  The authors should cite their source for the base map?  Not particularly useful as presented because it includes only about half of all sampling sites, gives no highlight of selected river systems, shows no lakes or dams, etc.  Presumably a DEM exists for this region but perhaps not at the resolution needed?  Many conservation organisations have better maps?  Even on a lower resolution map the authors could label their sampling sites while also emphasising the biodiversity importance of this region?

Figure 2.  Not sure what this figure shows us?  Would it offer a basis for comparison to another region in India or to another mountainous biodiverse region elsewhere?

Figures 3, 4 and 5.  All these figures need indication of the sampling sites within their respective river system and - if the authors want to focus on elevation - sort within each river by altitude?

Figure 6.  We need more information about the numerical basis for similarity-dissimilarity.  A user doesn't gain much useful information from this without designation of the rivers?  Or perhaps of elevation?

Figure 7.  Do these pictures come from this collection or from other sources?  I suspect ESSD can not publish them without attribution.  Do individual pictures associate with appropriate species in the database?

---

## Author Comment (AC2) · 24 Apr 2018

Reviewer Comments and Responses

Comment 1: The Pangaea link works very well. The authors basically repeat the tables from the manuscript as data files in the Pangaea archive. Because the Pangaea landing page correctly identifies the four data tables (tables 1, 3, 5 and 6) as tab-delimited files, the names of the files in the dataset folder should carry this designation. E.g. Arunkumar-etal_2015-T3.tsv rather than Arunkumar-etal_2015- T3.tab. R, Excel and other spreadsheets can easily ingest a .tsv file but those software packages will not recognize a .tab file. Once I renamed them I had no problem to open all the .tsv files.

[Figure]

Response 1: As per the reviewer comment the .tab file has been converted to .tsv file and attached separately. The ".tab" file format is already tsv-compatible, if you remove the "metadata header" from them (between the "/\*" and "\*/" tokens). For completeness of metadata and data when downloading it, we have to add the metadata to the files. In case you don't need them or your tool cannot process them, you may strip them away with a simple "sed"/"awk"/... command. There are also packages available to use PANGAEA data with several tools (Pan2Applic for ODV, PangaeaR for R, possibly also for Matlab) that will take care for you, while maintaining correct citation of the metadata.

Lines 13, 14. This sentence at the start of the abstract, about studying the freshwater fish from 2010 to 2013 can give a wrong impression. Some readers will assume that you studied these rivers in repetitive years, e.g. in 2010, again in 2011, out to 2013. In fact this study reports the outcome of a collection and identification process that covered 31 separate sampling sites that required 4 years (2010 to 2013) to complete. Nowhere in the documentation do we read about any repeat sampling. You should make very clear that you conducted one comprehensive sampling and assessment of each site in a process that required 4 years, 2010 to 2013.

Response 2: Yes the corrections has been incorporated in the manuscript as per the reviewer comment.

Comment 3: Lines 16, 17. These lines, about 64 species, some many orders, families, etc. repeat information from line 14. Remove the sentence in line 14? We do not need to see this information twice, so close together.

Response 3: The sentence has been removed.

Comment 4: Line 18. Plural of 'genus' should appear as 'genera' (as in line 17 above)?

Response 4: Yes the necessary corrections has been incorporated in the manuscript as per the reviewer comment.

Comment 5: Lines 23 to 26. The collection and identification of some fish species considered endangered or critically endangered probably represents an import contribution of this study. These fragmented and confused statements do not provide an adequate summary. The manuscript that follows probably needs an explicit section on endangered species and the significance of finding them in the Western Ghats. This abstract implies a "concise discussion" but that discussion never appears in the subsequent manuscript?

Response 5: Reviewer comment on concise discussion has been elaborated in the discussion part of the manuscript (Line no: 206 to 329).

Comment 6: Line 45, 46. The "Satpura" hypothesis? Specific to Indian or South Asian ecosystems or something more general. How and why did that hypothesis stimulate discussion on, for example, "endemism". Does this discussion have relevance to this paper?

Response 6: Comment on Satpura hypotheis by reviewer and its explanation has been discussed in the subsequent paragraph and reframed according to reviewer comment.

Comment 7: Lines 48 to 62, this paragraph about new species discoveries has relevance to the larger point about the Western Ghats as a location and source of unique biodiversity? We need to know the reason for this list of species names.

Response 7: The lines 48 to 62 describes about the species diversity of the southern Western Ghats. The list of species gives full views about the biodiversity status.

Comment 8: Line 65, 66. This combination of various types of nets has some minimum capture size? Larger organisms could avoid the nets? Please give users a sense of the size range captured by your sampling. What happens to any invertebrates? Discarded, recorded, ignored, or not captured?

Response 8: The mesh size of the fishing nets has been included in the manuscript. The other species were not disturbed.

Comment 9: Line 67. Five specimens from each species. This means you did some species identification in the field? Did you have a statistical basis for this sampling strategy? Or a valid logistical constraint? By focusing on repeatable numbers of species present, you have minimised the presence-absence question? The issue of absence of expected species does not arise in this manuscript, despite earlier mention of disturbance and invasives? Perhaps this discussion belongs in a subsequent research paper but those future researchers will need to know how to understand your data. Absence means zero specimens of a given species collected or fewer than 5 specimens collected?

Response 9: Minimized sampling is done to reduce the disturbance done to the diversity of species. Further species identification was done at the field level. Further the morphometric and meristic character analysis, a minimal no of specimens (5 no's) were transported to the laboratory. Further for I acknowledge that zero means that no specimens were recorded at that particular sampling sites.

Comment 10: Line 81, water samples collected "post-monsoon". We should have actual dates for all collection episodes, fish and water? Perhaps these exist in your database but they do not appear in any of the data shared here.

Response 10: The correction has been carried out in the manuscript. Comment 11: Lines 98, 99. Confusion here for the reader about primary sources of uncertainty. One source, identified here, involves conversion of information from "elementary" or "original" data sheets? Which feeds which? Which represents inputs to the database? Errors frequent or rare in these data translation processes? We need more clarification here. Later, in the summary, you should list for users all the known sources of uncertainty.

Response 11: The lines 98 and 99 has been removed.

Comment 12: Line 104. Present collection sites based on prior literature reports? Shouldn't we have this information, along with discussion of impact on reliability and
repeatability, earlier, in the methods section? If true, this opens the possibility of comparison of abundance, species presence- absence, etc., with earlier collections? We need to know if, where, and to what degree the present collections enable these historical comparisons. We do not need the comparisons themselves - those perhaps belong in a separate research paper - but we do need to know if the current collections enable such comparisons with prior collections. If not, why not?

Response 12: The present data expresses a vast variation in diversity to the earlier reports which has been elaborated in the discussion part in the manuscript.

Comment 13: Lines 104 to 112. This description of geomorphology and biodiversity of the Western Ghats belongs in the introduction?

Response 13: This description of geomorphology and biodiversity of the Western Ghats has been included in the introduction.

Comment 14: Line 114. We need a much better presentation of the sampling sites. Figure 1 shows only 16 out of 31 sites. If the authors want to show locations on a map, we need a better, more complete map. In Table 1 we get very useful information of sites by river system with elevation and lat lon for each site. But in Table 5 and 6 we lose that information. Those tables should still make clear the association of sites with rivers; you do not want us as users making mistakes in our assignment of site to river system. Table 1 does not seem to sort the site by elevation; somewhere we need that information. In the .tsv data file for table 1 I could sort by river system and then elevation (or forest type); better the authors should do this for all readers?

Response 14: Regarding the map, table 1, 5 & 6 the necessary corrections has been incorporated in the manuscript.

Comment 15: Line 119. Discussion of diversity, abundance and distribution by site, but not by river system, elevation, drainage area, proximity to human influence, presence or absence of hydroelectric dams, etc. Later in this results section the authors mention elevation, water temperature, water quality, lakes, etc. But here we don't get any sense of those factors for all sites or for each river system; we only see site-by-site lists.

Response 15: Based on the reviewer comments, the entire data has been organized.

Comment 16: Line 128. Species similarity was very 'low' rather than very "less"? The authors only hint at all the factors that might impact similarity, e.g. as plotted in Figure 6. Again we would need to see those sites according to their river systems rather than independently? Or, do the authors imply that site-to-site differences exceed river-to-river differences? We do not get a clear treatment of river vs site data and differences that would allow us to assess the validity of such similarities or differences. A clearer presentation of sites by river system, and of summary statistics of river vs river, or of the absence of significant differences, would help!

Response 16: The entire data has been organized based on the reviewer comments.

Comment 17: Line 138. IS: 10500 Permissible limits. This represents an India-wide water quality standard? We need a reference to it?

Response 17: The reference has been included in the manuscript as per reviewer comments.

Comment 18: Lines 137 to 154. This water quality discussion occurs almost entirely by site, not by river system. Why? Here and in Table 6, I feel surprised to see salinity. Conductivity I expect, and perhaps resistivity, but what do these very low values of 'salinity' represent and report? Do we have a definition of the salts involved? Not the typical seawater salts, presumably, so we must have a different freshwater definition of salinity? In other ESSD papers reporting mountain stream water quality, they typically do not report salinity. Because all these values lie below 0.5 ppt (one commonly-accepted definition of freshwater) we should consider them all to have negligible amounts of salts? If the authors do not make explicit use of the salinity data, we do not need them?

Response 18: As per the reviewer comments, the entire data on salinity has been removed.

Comment 19: Lines 156 to 210. After the water quality paragraph the authors provide four paragraphs on species appearance, diversity, abundance, etc. without clear conclusion and particularly without confirmation of Western Ghats as a biodiversity hotspot or as a 'refuge' for endangered species. Either we need less of species lists or we need more synthesis and assessment of the fundamental question: will these data allow and support discussion and conclusion about regional biodiversity and biological refuges or not - even if those discussions occur in other research papers or in other conservation fora. Readers need to get from the authors a sense of confidence on how to use these data! Unfortunately, from this somewhat confused and random discussion of physical and biological influences on habitats and species presence or abundance, we fail to get a clear understanding of the authors' confidence in their own data. We also learn that the Periyar river flows westward when earlier we read that this study focused on eastward-flowing rivers and about the existence of Periyar Lake. Certainly the map in Figure 1 and none of the text so far gave us any hints of lakes. Response 19: Yes the necessary corrections has been incorporated in the manuscript as per the reviewer comment.

Comment 20: Line 188: "moolavaigae"? Presumably this refers to a high-elevation location of Moolavaigae?

Response 20: Yes, moolavaigae refers to one of the high-elevation location among the Western Ghats where the Periyar River system originates its flow line no. 236- 238.

Comment 21: Line 212, 213, Summary. I do not understand the point of the first sentence about morphological variation related to micro- or macro-habitat?

Response 21: The morphological-based fish taxonomy is more inconclusive methods that explains the micro and macro habitat which may create much impact on the phenotypical variations among the fishes.
Comment 22: Lines 220 to 222. Again by individual sites, with no reference to river systems.

Response 22: Yes the necessary corrections has been elaborated in the manuscript in line no. 377- 396.

Comment 23: Line 223: The present study failed to convince this reader that altitude had any consistent eïñÃect. Probably more related to a weakness in the presentation rather than a weakness in the data, but in either case the paper has not shown us convincing data relating biodiversity to elevation. Response 23: As stated by the reviewer the changes has been carried out in the manuscript. Comment 24: Lines 224 to 232: Perhaps valid statements toward a positive "ecological spirit", but in fact from these data the authors have not guided us to conclusions about "sharp decline" (line 227) or about social pressures. Response 24: Yes the following changes has been carried out in the manuscript.

Comment 25: A good conclusion should briefly summarise the data, explicitly caution users about uncertainties and limitations of the data, and then outline both the present impacts (for conservation management) and need or intention for future monitoring or data gathering.

Response 25: Yes the necessary corrections has been elaborated in the manuscript in line no. 377- 396.

Comment 26: Table 1. 'Forest Type': these terms come from FAO or GBIF definitions or from an India Forest Classification scheme? Readers need to know how to relate this terminology to other data from other regions. 'Stream Order': hydrologists will understand this general mechanism to indicate stream branching but the authors should specify whether these represent standard Strahler stream order numbers or some other India-specific index of stream branching? 'Area' presumably represent catchment area above (upstream) of the sampling location. The authors should inform readers how they calculated this or from what source they extracted this information. Likewise for 'Volume', this presumably represents annual mean volume measured at some exit or drainage point of each stream and river? Again, readers need to know where this information comes from. In this table, 'Mean Velocity' represents a code referenced in a footnote, and not an absolute value? The authors could reduce confusion by using the actual codes 'slow', 'moderate', 'very fast', etc? Response 26: The classification of the Forest Types were given based on the India Forest Classification scheme. The link about the classification are attached. http://www.sikkimforest.gov.in/docs/Forestry/Vegetation%20Types.pdf http://www.biologydiscussion.com/forest/5-types-of-forests-found-in-india-explained/6940 The rest of the corrections regarding the stream order, area, volume, mean velocity has been corrected in the manuscript. The following article gives the detail information regarding the parameters followed. Armantrout, N.B. 1990. Aquatic habitat inventory. Bureau of Land Management, Eugene District, U.S.A. pp. 32.

Comment 27: Table 2. Comprehensive species list but the IUCN codes in column 4 do NOT match the descriptions in the footnote. For example, the footnote does not define LC as it appears in the Table while 'LRnt' from the footnote never appears in the Table. Response 27: Yes the necessary corrections has been incorporated in the manuscript as per the reviewer comment.

Comment 28: Table 3. We need these data explicitly organized by river system and perhaps sorted in order of elevation within each river system. With efiort a user can establish these distinctions and filter by elevation by using the .tsv file but the authors need these improvements in Table 3 in order to support their discussion about, for example, biodiversity and altitude? Response 28: Yes the necessary corrections has been incorporated in the manuscript as per the reviewer comment.

Comment 29: Table 5. Species by generic number code vs. site by similar generic number code. Inclusion of actual species names (as genus.sp) and clearer organization by river system would greatly increase the utility and information content of this

Table!

Response 29: Yes the necessary corrections has been incorporated in the manuscript as per the reviewer comment. Comment 30: Table 6. Again, organize this by river system and sort by elevation?

Response 30: The table has been organized based on river systems and the elevation has been detailed in table 1 & 3.

Comment 31: Figure 1. The authors should cite their source for the base map? Not particularly useful as presented because it includes only about half of all sampling sites, gives no highlight of selected river systems, shows no lakes or dams, etc. Presumably a DEM exists for this region but perhaps not at the resolution needed? Many conservation organizations have better maps? Even on a lower resolution map the authors could label their sampling sites while also emphasizing the biodiversity importance of this region? Response 31: Yes, the map has been plotted using the base map provided by the gramin gps instrument. Comment 32: Figure 2. Not sure what this figure shows us? Would it oïñÃer a basis for comparison to another region in India or to another mountainous biodiverse region elsewhere?

Response 32: Regarding the figure 2 the author explains about the six different orders collected from Western Ghats of which the order Cypriniformes are dominant.

Comment 33: Figures 3, 4 and 5. All these figures need indication of the sampling sites within their respective river system and - if the authors want to focus on elevation - sort within each river by altitude? Response 33: The sampling sites within the river system were differentiated and the elevation details were incorporated in the table 3. Comment 34: Figure 6. We need more information about the numerical basis for similarity-dissimilarity. A user doesn't gain much useful information from this without designation of the rivers? Or perhaps of elevation? Response 34: Yes, the reviewer has commented on the Figure 6, in regard to that the author has plotted a cluster dendogram using all the water quality, diversity parameters and habitat characters of

the six river systems to find out the dissimilarity among the sampling sites to prove that the habitat characters plays a vital role in the diversity of species among the six river systems.

Comment 35: Figure 7. Do these pictures come from this collection or from other sources? I suspect ESSD cannot publish them without attribution. Do individual pictures associate with appropriate species in the database? Response 35: Yes, these pictures are original taken by the authors from the streams and rivers of Western Ghats. Further the pictures in figure 7 and the species list in table 2 is available in the database.

Please also note the supplement to this comment:
https://www.earth-syst-sci-data-discuss.net/essd-2017-118/essd-2017-118-AC2-supplement.pdf

**Supplement:**

**FRESHWATER FISH FAUNA OF RIVERS OF SOUTHERN WESTERN GHATS, INDIA**

Anbu Aravazhi Arunkumar[1], Arunachalam Manimekalan[2]

[1]Department of Biotechnology, Karpagam Academy of Higher Education, Coimbatore 641 021.

Tamil Nadu, India

[2]Department of Environmental Sciences, Biodiversity and Molecular Lab, Bharathiar University, Coimbatore 641 046, Tamil Nadu, India.

*Correspondence to*: Anbu Aravazhi Arunkumar (anbu.arunkumar@gmail.com)
https://doi.org/10.1594/PANGAEA.882214

**Abstract.** This paper provides information on the diversity of freshwater fish fauna of six river systems of Southern Western Ghats. The fishes were collected using cast net, dip net, gill net and drag net from various streams and rivers. There are about 31 sites in which a total of 64 species belonging to 6 orders, 14 families and 31 genera were recorded. Among them the order Cypriniformes was dominant with 3 families' 18 genera and 49 species (76.6%) compared to other orders. Principal component analysis and cluster analysis were performed to express the contribution of the variables and its influence on the species diversity. Interestingly, of the 31 sites Thunakadavu stream, Gulithuraipatti, Athirappalli, Naduthotam, Nadathittu, Mullaithodu, Thonanthikla, Noolpuzha and Sinnaru exhibited high variations in species diversity. Nearly fifteen species were found to be threatened to the Western Ghats. *Garra periyarensis* and *Cirrhinus cirrhosus* are known to be vulnerable and *Hemibagrus punctatus* is Critically Endangered because of various anthropogenic activities. The study clearly indicates that certain timely measures has to be taken immediately to protect the fishes in Southern Western Ghats.

**Keywords:** Southern Western Ghats, Water Quality, Species Diversity, Endemic, threats, Conservation.

**1. INTRODUCTION**

[revised manuscript text omitted]

**2.3 Interpretative analysis**

To quantify species diversity, for the purposes of comparison, a number of indices have been followed. To
measure the species diversity (H) the most widely used Shannon index (Shannon and Weaver, 1949), Evenness
index (E) (Pielou, 1975), and Dominance index (D) (Simpson, 1949) were used. Similarity coefficients of the
fish community were calculated by using Jaccard index (Southwood, 1978). The species abundance and their
relative frequencies were subjected to cluster analysis, a complete linkage cluster dendrogram was drawn based
on Pearson correlation. The contribution of the variables and its influence for the species diversity has been
analyzed using Principal Component Analysis (Wills, 2005). The above statistical analyses were performed
using SPSS (version 21), XLSTAT, and Biodiversity Pro software's.

**3.  RESULTS AND DISCUSSION**

Fish Fauna were surveyed from the streams and rivers of Southern Western Ghats. Collection sites were selected
based on the earlier faunal distribution published in literature. The Western Ghats is a mountain range that runs
almost parallel to the western coast of Indian peninsula. The study sites and its characteristics are recorded and
presented in Table 1 and Fig 1, 1a. In the present work a total of 31 sites of six river systems of Southern
Western Ghats were studied of which a total of 64 species belonging to 6 orders, 14 families and 31 genera were
recorded (Table. 2). Among them the order Cypriniformes was dominant with 3 families' 18 genera and 49
species (76.6%) compared to other orders. (Fig.2, Fig.7).

**3.1 Fish Species Density, Abundance, and Distribution**

Among the 31 sites high species diversity was recorded at Sinnaru of Cauvery River system (H'-
1.268) and low diversity was recorded at Thunakadavu tunnel of Chalakudy River System recorded (H'- 0.357)
(Table: 3, Fig: 3). The maximum species diversity was recorded in Sinnaru of Cauvery River system (S – 21)
and the minimum was recorded at Puliyarkutti 3[rd] bridge and Thunakadavu tunnel  of Chalakudy River System
and Sorrakottaodai of Periyar river system (S – 3), (Table: 3, Fig: 4). The maximum species abundance 152 was
recorded at Naduthottam of Periyar river system and lowest 16 was recorded at Sorrakottaodai of Periyar river
system and Belikoondu of Cauvery river system (Table: 3, Fig: 5). The maximum dominance index (D - 21.346)
was recorded at Sinnaru of Cauvery river system and lowest (D- 2.121) was recorded at Thunakadavu tunnel  of
Chalakudy river system (Table: 3).

**3.2  Species composition**

Species similarity between the sites was very low among 31 sites of six river systems. Cluster analysis exhibited
similar species composition between the sites (Table: 4, Fig: 6). Totally 5 clusters were grouped which clearly
demonstrate the similarity of species composition among the sites. The cluster group separation is based on the
following reasons – 1. Due to the rare species forms, 2. Due to low water temperature and 3. Prevalent human
disturbances.

**3.3  Water Quality:**

Water Quality parameters were recorded and presented in table 2.6. It is found that the parameters
value lies in the IS: 10500 Permissible limits. (Table: 6) (BIS 2012). The selected sites of Western Ghats has
water pH ranging from 6.5 to 8.5. A pH of 9 was recorded at Kadapilliyarthittu of Cauvery river system and 7.2
was recorded at various sites like Anjurily, Athirapalli, Urilikal. Minimum conductivity value 27.8mS was
recorded in Chalakudy river system and maximum conductivity value 85.2mS recorded in Noolpuzha of Nugu
river system. Total dissolved solids (TDS) are a measure of inorganic salts dissolved in water. This dissolved
solid comes from both natural and human sources. Mitchell and Stapp (1992) have suggested that changes in
TDS concentrations can be harmful. If TDS concentrations are too high or too low, the population of aquatic life
can be limited. Thenkasithodu of Periyar river system witnessed a low value of TDS content as 13.7 mg/l and
Urilikal of Chalakudy river system recorded a high value of TDS as 51.9mg/l. A minimum Resistivity value of
2.58 was measured at Kadapilliyarthittu of Cauvery river sysyem and a maximum 45.6 was measured at
Thenkasithodu of Periyar river system. A high level of DO of 6.11mg/l was recorded at Thenkasithodu of
Periyar river system and low DO of 0.63 mg/l was recorded at Belikoondu of Cauvery river system. Arunkumar
*et* al., (2015) recommended that the lowest DO recorded at sampling sites is due to organic-rich domestic waste
let into the river by the tourists in the river system. Maximum water temperature (33.6°C) was recorded at
Pillapara of Chalakudy river system and minimum water temperature (18.9°C) was noted at Thenkasithodu
Periyar river system.
**3.4 Habitat Structure**
Stream habitat was measured in dimensions like length, width, depth, substratum and current. Large
proportions (> 50%) of the habitat sampled included very shallow water (< 1 centimeter). Typically, such areas
are not habitable by fishes and most fish concentrate in dispersed pools indicating that habitat measures in up-
stream areas should be restricted to the pools themselves (Gorman, 1978).
**3.5 Substratum Types**
For the present study, the fish species diversity, habitat quality assessments of the river systems have
been taken as the major criteria. The results exhibit that the study area is well flourished with flora and fauna. It
proves that habitat provides the perfect level of food, shelter suitable for the fishes and other aquatic organisms.
The habitat assessment of the study area says that there are four habitat types (pool, riffle, run and glide) with
six substratum type (Fine sand, debris, Silt, Bedrocks, Gravel, Rubbles and boulders). The shore line is also
sandy border rigid with rocks which makes up a good habitat for the aquatic organisms. Moreover the water
quality, substratum type and vegetation provide a good and healthy habitat and high food resource availability which plays a major key role for species diversity. The river habitat is utilized by the tribal people for catching the fishes for their source of protein food. In the present study, in regard to the substratum types like Rubble and

Boulders were the dominant with 80% in Mullaithodu of Periyar river system. Anjurily of Periyar river system gravel was the dominant substratum representing 70%. Moreover substratum types like sand and silt are equally represented in all the study sites. Debris is the biological matter that occupies the stream habitat as a major part in providing good shelter and feeding habitat for the fishes. Mostly the bottom feeders like *Garra, Nemachelius,*

*Travanchoria* use these debris and bed rock substratum as their habitat in a total stream channel with all other substratum types. Nadathittu of Cauvery River, Naduthotam of Periyar River, Kovaikutralam of Bhavani River and Thunakadavu of Chalakudy river system have their base substratum as natural bedrock, which provides them a strong rigid bottom.

**S**tream width and volume were high at Belikoondu of Cauvery river system (80 m, 80000 m$^3$) followed by Nadathittu (70m, 42000m$^3$), Kadapilliyarthittu (75m, 11250m$^3$), Kallampalayam (13m, 10400m$^3$),

Noolpuzha (25m, 10250m$^3$). The lowest stream width and volume were recorded at Thellikal (4m, 400m$^3$).

Among the 31 sites very fast flowing water was noted at Nellithurai, Thunakadavu tunnel and Belikoondu. Fast flow and Moderate flow water was noted in most of all the river systems. Slow flow of water in the channels was recorded at Thenkasithodu, Kadapilliyarthittu, Oorpannikaham and Urilikal.

**3.6 Ecological Structures Influence Characterizations**

Principal component analysis was used to illustrate the influence of the variables and its importance for the ecological structure of the river system and the fish species. The various habitat characteristics like water quality, channel morphology, and the substratum type influencing the species distribution. Factors like Altitude (6.940), Area (21.449) and Volume (58.428) influence the species diversity (Table.7). All other characters play a supportive role to express the variations among the study sites. Based on the contributions study sites like

Belikoondu, Kallampalayam, Sorrakottaodai, Anjurily, Thenkasithodu, Belemeenthurai, Kovaikutralam,

Naduthotam, Nadathittu, Kadapilliyarthittu and Sinnaru exhibits more variations. The results obtained concludes that altitude plays a major role in species diversity and species abundance, which supports the proposed the theory that diversity changes with altitude on mountain sides, being lowest at higher elevations (Colinvaux,

1930). The present finding supported the above theory as the results expressed that species diversity and abundance is low at high altitudes. Among the 31 sites, high species diversity was recorded at Sinnaru of

Cauvery River system (H'- 1.268) because of the altitude, area of the channel and the volume of flow as well.

The maximum species diversity was recorded at Sinnaru of Cauvery River system (S – 21), due to the channel flow, altitude and the submerged substratum types with muddy water flow. The maximum species abundance

152 was recorded at Naduthotam of Periyar River system due to the low area of the channel and the maximum percentage of the rocky boulder substratum. The maximum dominance of species (D – 21.34) was recorded at

Sinnaru of Cauvery River system influenced by the vast channel area. Rest of the sites was low due to the less percentage of influence made by the habitat structures.

Rajan (1955) has studied the fishes of Moyar river system and has reported 48 species. Manimekalan (1998) has reported 38 species form Mudumalai wildlife sanctuary. Manimekalan has stated that species like

*Labeo dero, Puntius mudumaliensis, Schimatorhynchus nukta, Danio neilgherriensis, Crossochelius latius*

*latius, Clarias dayi, Gambusia affinis* were restricted to Moyar river system. Also *Clarias dayi* a critically endangered species has been recorded by Manimekalan (2002). *Puntius carnaticus* and *Danio aequipinnatus* was recorded as common species of Moyar river system. Rajan (1955) and Mukerjii (1931) has studied the headwaters of Bhavani river and reported species like *Travancoria elangata, Barilius canarensis, Rasbora*

*caveri, Garra menoni, Silurus wynaadensis* were restricted to the river system. *Puntius filamentosus, Puntius*

*melanampyx, Puntius carnaticus, Barilius gatensis, Danio aequipinnatus, Rasbora daniconius* were very common in Bhavani River System. Arunkumkar *et al.*, (2015) has recorded nearly 37 species from Cauvery river system. Among several fish species, the only *Garra gotyla stenorhynchus* is reordered as one of the endangered species in Grand Anicut Cauvery, which is locally consumed (Murthy *et al.*, 2015). But *Garra*

*gotyla stenorhynchus* is still under least concern status of IUCN.

Silas (1951) in his faunal account discusses the extension of range of *Salmostoma acinaces* (*Chela*

*argentea* Day), Barbodes carnaticus (*Barbus* (*Puntius*) *carnaticus*), *Osteochilus* (*Osteochilichthys*) *thomassi* and

*Batasio travancoria* and lists 2 endemic species described by Herre viz. *Homoloptera Montana* and

*Glyptothorax housei.* Silas further reported 5 species from the Cochin part of the anamalai hills viz. *Barilius*

*bakeri, Puntius denisoni, Travancoria jonesi, Noemacheilus triangularis* and *Batasio travancoria. Punitus*

*bimaculatus* earlier considered as a juvenile of *Puntius dorsalis* has been collected from Anamalai hills.

Interestingly this species is found to be the most dominant *Puntius* species in the hill ranges of the Eastern Ghats especially Javadi hills. *Puntius punctatus* earlier considered as a synonym of *Punitus ticto* has been kept as a separate species and both these species have been collected from Anamalai hills (Menon, 1999).

Diversity in the Anamalais is very high except for a few areas such as the Aliyar river basin.  The lack of diversity in the Aliyar river basin is due to the fact that most of the streams in the area are non-perennial and are prone to disturbance/contamination by the local tribal people. This diversity is attributed to the controlled fishing activity by locals and protection by Forest officials. The physical environment like forest vegetation, riparian vegetation, water temperature, habitat type, and in-stream cover (which provide hiding places for fish)

play a major role in species diversity.  The Periyar River originates near moolavaigae and reaches the Mullai

Periyar Reservoir located in the premises of Periyar Tiger Reserve which is one of the biodiversity rich zone in

Southern Western Ghats (Silas 1950, 1952; Kurup *et al.,* 2004). Earliest studies on the fish fauna of the PTR

dates back to 1948 when Chacko (1948) listed 35 species from the Periyar Lake, including the critically endangered small scaled *Schizothoracin Lepidopygopsis typus.* Later Menon and Remadevi (1995) described

Hypselobarbus *kurali* from streams adjoining the Periyar river raising the total number of fish species to 38. In the present study 64 species were collected from 31 study sites of six river systems of southern Western Ghats.

Species like *Puntius melanampyx, Puntius carnaticus, Puntius amphibious, Puntius fasciatus, Puntius*

*mahecola, Devario aequipinnatus, Garra mullya, Travancoria jonesi, Nemacheilus guntheri* were commonly found in all the six river systems (Fig:7).

Smith has stated that habitat selection of the fishes is influenced by the body structure, food and shelter and by physiological process. Moreover the fish analyses the characters of the rivers and streams and further they respond to the characters and helps themselves for the survival of the fittest. Hence it is reliable that the

Micro and Macro habitat plays a key role in the morphology and physiological characters and modifications of the species. The fish prefers the habitat based on the nature of the rivers or stream substratum type where the muddy bottom with debris is records for high species richness of the bottom feeders. Odum (1945) well stated that the flow of the water in the channel is an important factor controlling the distribution of fishes, the species like *Barilius, Hypselobarbus, Puntius, Travancoria, Rasbora* and *Tor* prefers fast flow. The nature of the substratum and the flow rate seem to be more or less closely interrelated in governing the distribution of the fishes. This induces the dominance of the cyprinid species to be well flourished in all the river systems, of the

Southern Western Ghats. It is clear that Ecological structure plays a key role in representing River Systems of

Southern Western Ghats which is flourished with rich species diversity and abundance.

Conservation of India's vast and diverse aquatic genetic resources is essential to maintain ecological as well as socio-economic equilibrium. Fisheries and aquaculture have a promising role to play in social development by providing nutritional security for the Indian population and contributing to the economic advancement of farmers and fishery workers. The concept of fish conservation is not new to India. The fishing was prohibited during the third Chatturmass (July- October) to protect the pre-spawning brood stock and young ones. King Ashoka's prohibition period extends up to the middle of November. Renowned fisheries taxonomist

Francis Day drew the attention of the Government of India to large scale slaughter of fish fry and fingerlings and pleaded urgent conservation measures. After persistent pressure, the Indian Fisheries Act was enacted in

1897. The destructive fishing methods, creation of 'fixed engines' (dams, weirs etc.) for catching fish and use of small sized nets were banned by the law. The main threats impacting freshwater biodiversity in the Western

Ghats include pollution (urban and domestic pollution ranking as the worst threats followed by agricultural and industrial sources of pollution), residential and commercial development, dams and other natural system modifications, invasive species, agriculture and aquaculture, energy production and mining (IUCN). The anthropogenic perturbations to fresh water systems over the past years have escalated to enormous proportions and it is estimated that about 3000 species will become extinct within the next 20 to 30 years (Das, 1994;

Prasad, 2010).

The threat to the endangered fish species from our aquatic ecosystem can be minimized by employing both preventive and protective measures. The preventive measures may include removal of causative factors and provision of suitable legislation. The protective measures would include identification of suitable areas to declare as sanctuaries and develop new technologies for the protection of the genetic resources of threatened and vulnerable fish species. Keeping this in view, the present investigation highlights some of the main causative factors of decline *Tor* species and some remedial measures for preserving the fish population. Degradation of aquatic systems, indiscriminate fishing of brood fish and juveniles, anthropogenic intervention, use of explosives, poisons and intrusion of exotic species are the major possible factors noticed in the present study which causes the depletion of fish population in the study area. Several authors have observed that fish population has recorded a sharp decline in Indian rivers due to the indiscriminate fishing of brood stock and juveniles, fast degradation of aquatic ecosystems and construction of dams, barrages, weirs, etc. Many factors have been noticed during the present study which affects the fish population adversely. Indiscriminate fishing of brood fish and juveniles, use of explosives, poisons and electrocution are some of the major possible factors the causes of depletion of fish in Indian waters.

The tribal fishermen have more preference towards fish species because of its large size and medicinal properties. The use of different types of plant products for fishing was observed among the tribes, which will kill all the fishes including young ones. *Croton tiglium* L., *Gnidia glauca* (Fresen) Glig., *Acasia instia, Acasia*

*torta, Hynocarpus pantandra* are some plants which can be used as fish poison for catching fishes. The parts of the plant (leaves, stem, bark, fruits and seeds) and the whole plants are used as fish poison. This method can be applicable only in stagnant water which leads to mass poisoning (Ambili, 2013). Dynamiting is a common practice seen among the tribals and its frequent used in stagnant rock pools and deep water body. In this method all the fishes available, from juveniles to adults, in the spot will be affected. Dynamiting is practiced by the tourists who visit the places illegally. Use of explosives, poisoning, electrocution and use of small sized nets etc. are some other fishing methods which affects the population adversely (Ambili, 2013).

Use of Copper Sulphate is a destructive method of fishing leads to mass poisoning of fish population. Irulas, Kurumbas and Mudugar are the tribal settlement in the Attapadi region on the banks of Bhavani river. They use the cast nets, gill nets and bamboo traps (Kooda) for fishing. Indiscriminate fishing in the Bharathapuzha made a large decline of *Tor* population. Sarkar and Srivastava (2000) noticed that because of increased anthropogenic activities, the two main species namely *Tor. putitora* and *Tor. tor* are listed under the category of endangered species and facing the high risk of extinction in the wild. Due to the proximity of human settlement, aquatic ecosystems are relatively more exposed to human influences and interventions. Besides, the industrial and urban development has altered the aquatic environment. Overfishing at various stages of life-cycle have observed more in human settlement area and this causes the spectacular changes in the environment affecting the fishery resources. Polluting the water body is also one main factor which causes the declining of the ichthyofaunal diversity (Ambili, 2013).

The pristine riverine systems along the Western Ghats have been tampered with anthropogenic activities such as dam constructions and road building which have affected the ecology and habitat of these fishes. The tourist resorts started down to Athirappally on the bank of Chalakudy river are altering the habitat by many ways. Sholayar Hydro Electric Project and Peringalkuttu Hydro Electric Project are the Hydro electric projects on the Chalakudy river. There are about 7 dams built on the river. Peringalkuthu Dam of this river prevents the local migration of *Tor* from the lower to upper stretches of the river. There are 11 reservoirs in Bharathapuzha river and Malampuzha dam is the largest one. Neyyar dam is located in the Neyyar river and the Idukki dam is located in Periyar river. The construction of dams also resulted in less water flow and affects the migration of fishes. Food availability is an important factor for the existence of fish species. MacDonald (1948) noted that Mahseer is an intermittent feeder. The vegetative matter, benthic diatoms, molluscan shells, crabs, insects, small fishes, different types of seeds and fruits have been recorded from the stomach contents of *Tor* (Dinesh *et al.*, 2010). The availability of these items varies considerably during different seasons which cause the mortality of young ones. The disruptions in the food chain also affect the species adversely. The deforestation rate all along the Western Ghats is so high and the forest areas are being transformed into agriculture practices. This type of practices was seen on the bank of most of the rivers in Kerala like Chaliyar, Sholayar, Chalakudy, Kabini, Bhavani, Periyar and Kallada river systems. Cultivation of Musa, Paddy, Cardomom, Ginger and Tea plantations are observed to be the major ones. The pesticides used in these areas are penetrating to the river system and severely affect the aquatic organism like insects, diatoms, vegetation such as phytoplanktons and even the small fishes (Ambili, 2013).

Fish population is declining rapidly hence the following immediate conservation measures will help to conserve this precious species. Awareness among the tribes is more important for conservation of fish species. Awareness can be made about the impact of using chemicals for mass poisoning, dynamiting  for catching fish, avoid fishing during breeding seasons and the use of poisonous plant products for mass poisoning. Student, social workers, fishermen and local people should be educated about the importance of conservation of fish fauna in their area so that they can make awareness among the nearby people. More exclusive projects should be
started with the co-operation of local people and students for protecting the fish population. Action can be taken
to change the fishing profession of those who only depend on fishing for their livihood which will help to reduce
the fishing pressure.

In order to conserve the fish genetic resources and provide adequate living space, shelter and habitat for
valuable threatened fishes, certain areas can be declared as fish sanctuary like National Parks and Wildlife
sanctuaries. Menon *et al.,* (2000) suggested that suitable segments of the rivers with fish species should be
identified for establishment of 'fish sanctuaries' and such sanctuaries must be heavily stocked every year with
fish fingerlings. There are two fish sanctuaries protecting the *Tor* species as a part of their religious customs,
they are Aruvikara (Neyyar river) and Kulathupuzha (Kallada) in Kerala. The upstream part of Chalakudy river,
Karimpuzha and Manjeeri region of Chaliyar river, a part of Bhavani up to Thavalam (Attappadi region) and
selected stretches of Periyar river can be declared as fish sanctuaries. Ambili et al., (2014) has reported the
presence of three species *Tor khudree, Tor malabaricus and Tor mussullah-* in the River Chaliyar. Long
stretches of Cauvery river is a fish sanctuary where the Karnataka Forest Deparment (Wildlife) has leased out 14
miles of the river Cauvery to Wildlife Association of South India (WASI), which is now protecting the wildlife
including fishes with more care.

Captive breeding is widely used throughout the world for a variety of endangered animals including
fish (Maitland and Evans, 1994; Keshavanath *et al.,* 2006). It could be an important 'last resort' measures for
endangered and endemic species, which may otherwise become extinct in the wild (Reid, 1990). Fishery
Departments should take steps for breeding and caring of the endangered *Tor* species. In Kerala near Pookkode
Lake and Sholayar dam, procedures for culturing the *Tor* species were attempted in hatcheries, but they could
not succeed. Collection of matured brooders from interior of the forest and maintenance of water temperature
are the two major problems came across. Now studies are going on to compensate the reasons for failures in *Tor*
breeding. Gene banks can hold live animals or cryopreserved gametes. Gene banks can be considered as a last
line of defence against species extinction. A live gene bank contributes to delisting of threatened species by
captive breeding and restocking in species-specific recovery programmes. Such gene banks can contribute to
recovery and utilization of genetic diversity and can be used in conservation programmes (e.g., NBFGR, India
and the World Fisheries Trust, Canada) and genetic enhancement (e.g., salmon in Norway and common carp in
Hungary) (Lakra *et al.*, 2007). A mini gene bank with the milt of *T. putitora* and *T. khudree* has been established
by NBFGR (Ponniah *et al.*, 1999a; 1999b). In India *Tor* spermatozoa cryopreservation protocols have been
developed by several workers (Basavaraja and Hedge, 2004, 2005; Patil and Lakra, 2005). Fish sperm
cryopreservation requires the development of species-specific protocols (Lakra *et al.*, 2006). Cryopreservation
of germplasm is a very good *ex situ* strategy to conserve existing allelic diversity for future use. This technique
may help to provide gametes for artificial propagation programmes in the off seasons also. Universities and
Research institutes should be taken care of the cryopreservation and captive breeding of *Tor* species. Re-
introduction is very essential, than the introduction, for the conservation native species. Introduction can never
neutralise the problem of depletion of species. While, re-introduction (collection and protection of wild /native
fishes and introduce them in to the rivers) can support a lot towards the conservation of native species.
Introduction of *Tor* species in the rivers of Kerala from the other region or other river systems are making more confusions and taxonomic ambiguities and sometimes people wrongly quoting for supporting this evidence for
Satpura Hypothesis (Kumar and Kurup, 2004).

Monitoring and documentation of fish stocks are significantly important to carry out regular reviews on
the distribution and status of all fish species and will be possible by maintaining the registers (Koljonen and
Nyberg, 1991). The documentation of genetic resources for aquaculture is also the part of the coverage of Fish
Base (Froese and Pauly, 2013). The comprehensive listing of fish species distribution and continuous
monitoring of the fish species is the most critical need of protection.

**361    4. SUMMARY AND CONCLUSION**

The morphological-based fish taxonomy is more inconclusive methods that explains the micro and
macro habitat which may create much impact on the phenotypical variations among the fishes. In the present
study, the fishes were collected from various river systems of Southern Western Ghats like Bhavani, Moyar,
Chalakudy, Periyar, Cauvery and Kabini by using different mesh size of gill nets, cast net and dip net. A total of
31 sites of six river systems of Southern Western Ghats were studied in which a total of 64 species belonging to
6 orders, 14 families and 31 genera were recorded. Cypriniformes was the most dominant order with 3 families
18 genus and 49 species (76.6%) compared to other orders. Interestingly, the sites like Thunakadavu stream,
Gulithuraipatti, Athirappalli, Naduthotam, Nadathittu, Mullaithodu, Thonanthikla, Noolpuzha and Sinnaru
revealed high species diversity. The results indicated that the species from Southern Western Ghats have an
ambiguity taxonomy among the fish communities. The data analyses suggested that species like *P. melanampyx,*
*P. carnaticus, P.amphibious, P. fasciatus, P. mahecola* were found to be the dominant species in the locations
considered. Among the 31 sites, maximum diversity (H'- 1.268) was recorded at Sinnaru (altitude – 225) of
Cauvery River system and minimum diversity (H'- 0.73) was recorded at Urilikal (altitude – 3238) of
Chalakudy River system. The present finding supported the Colinvaux theory which expresses diversity changes
with regards to elevations. The nature of ecosystem and the vegetative forest which prevails along the river
systems of Southern Western Ghats are more suitable habitat for fishes. Many threats like use of explosives,
poisoning and fishing of juveniles are reported against the existence of the fishes from the rivers of Southern
Western Ghats. Hence, an urgent attention is needed to create awareness among local communities about the
importance of the stream habitat and its fish diversity, for conserving these important resources for future
generations.

**383    5.  ACKNOWLEDGEMENT**

The authors gratefully acknowledge facilities provided by the Department of Environmental Sciences,
Biodiversity and Molecular Laboratory, Bharathiar University.

[revised manuscript text omitted]

| Cluster no | Cluster between | Study sites |
|---|---|---|
| 1 | 1 - 4 | Thunakadavu stream, Baghapallam, Kallampalayam, Thunakadavu tunnel |
| 2 | 5 -7 | Thenmudiparai, Orukomban range, Gulithuraipatti |
| 3 | 8 - 28 | Melaparai, Valukuparai, Belemeenthurai, Anjurily, Oorpannikaham, Nellithurai, Belikoondu, Kadapilliyarthittu, Sorrakottaodai, Puliyarkutti 3$^{rd}$ bridge, Mullaithodu, Kovaikutralam falls, Puliyarkutti 8$^{th}$ bridge, Sinnaru, Nadathittu, Thonanthikla, Thellikal, Pillapara, Athirapalli, Noolpuzha, Ummikuppamthodu |
| 4 | 29 | Naduthotam |
| 5 | 30 | Thenkasithodu |
| 6 | 31 | Urilikal |

**Table 5: Distribution and abundance of fishes of six river systems**

| Species | 1 | 2 | 3 | 4 | 5 | 6 | 7 | 8 | 9 | 10 | 11 | 12 | 13 | 14 | 15 | 16 | 17 | 18 | 19 | 20 | 21 | 22 | 23 | 24 | 25 | 26 | 27 | 28 | 29 | 30 | 31 |
|---|---|---|---|---|---|---|---|---|---|---|---|---|---|---|---|---|---|---|---|---|---|---|---|---|---|---|---|---|---|---|---|
| *Puntius melanampyx* | | | | 4 | 6 | 12 | 7 | 10 | 5 | 11 | | 32 | 5 | 2 | 5 | | | 2 | 2 | | | 5 | 4 | 5 | 2 | | | | | | 10 |
| *Puntius carnaticus* | 1 | 1 | 5 | | | | 1 | | 2 | 6 | | | | | | | | | | | | | | | | 4 | | 9 | 5 | 2 | |
| *Puntius amphibius* | 1 | 2 | | | | | 1 | | | | | | | | | | | | | | | | | | | | | | 2 | | |
| *Haludaria fasciatus* | | | | | | | | 5 | 10 | 5 | | 13 | | | 15 | | 1 | 1 | | 2 | | 4 | 5 | | | | | | | | 15 |
| *Dawlinsia filamentosus* | 1 | | | | | | | | | | | | | | | | | | | | | | | | | | | 5 | 2 | 5 | |
| *Puntius sarana sarana* | 15 | 1 | | | | | | | | | | | | | | 10 | | | | | | | | | | | | | 6 | | |
| *Puntius dorsalis* | 1 | 1 | | | | | | | | | | | | | | | | | | | | | | | | | | | | | |
| *Puntius chola* | 7 | 15 | | | | | | | | | | | | | | | | | | | | | | | | | | | | | |
| *Puntius sophore* | | | | | | | | | | | | 11 | | | | | | | | | | | | | | | | | | | |
| *Eechathalakenda ophicephalus* | | | | | | | | | | | | | | | | | | | | 19 | 26 | | | | | | | | | | |
| *Puntius mahecola* | | | | | | | | | | | 25 | | 8 | 5 | | 5 | | | | | | | | | | 4 | 3 | 3 | | | |
| *Pethia conconius* | | | | | | | | | | | | | | | | | | | | | | | | | | 10 | 1 | 3 | 4 | | |
| *Sahyadria denisonii* | | | | | | | | | | | | | 5 | 5 | | | | | | | | | | | | | | | | | |
| *Sahyadria chalakudiensis* | | | | | | | | | | | | | 1 | 1 | | | | | | | | | | | | | | | | | |
| *Puntius sarana spirulus* | | | | | | | | | | | | | | | | | | | | 3 | | | | | | | | | | | |
| *Puntius bimaculatus* | | | | | | | | | | | | 10 | | | | | | | | | | | | | 2 | | | | | | 10 |
| *Pethia ticto* | | 1 | | | | | | | | | | | | | | | | | | | | | | | | | | | | | |

| *Species* |  |  |  |  |  |  |  |  |  |  |  |  |  |  |  |  |  |  |  |  |  |  |  |  |  |  |  |
|---|---|---|---|---|---|---|---|---|---|---|---|---|---|---|---|---|---|---|---|---|---|---|---|---|---|---|---|
| *Cirrhinus cirrhosus* |  |  |  |  |  |  |  |  |  |  |  |  |  |  |  |  |  |  |  |  |  |  |  |  | 6 | 2 |  |
| *Skymatorynchus nukta* |  |  |  |  |  |  |  |  |  |  |  |  |  | 2 |  |  |  |  |  |  |  |  |  |  | 3 |  | 5 |
| *Labeo boggut* |  |  |  |  |  |  |  |  |  |  |  |  |  |  |  |  |  |  |  |  |  |  |  |  |  | 1 |  |
| *Labeo kontius* |  |  |  |  |  |  |  |  |  |  |  |  |  |  |  |  |  |  |  |  |  |  |  |  |  | 2 |  |
| *Labeo ariza* |  |  |  |  |  |  |  |  |  |  |  |  |  |  |  |  |  |  |  |  |  |  |  | 1 | 1 | 2 |  |
| *Labeo calbasu* |  |  |  |  |  |  |  |  |  |  |  |  |  |  |  |  |  |  |  |  |  |  |  |  | 1 | 1 |  |
| *Labeo boga* |  |  | 2 |  |  |  |  |  |  |  |  |  |  | 1 |  |  |  |  |  |  |  |  |  |  |  |  |  |
| *Hypsilobarbus curmuca* |  |  |  |  |  |  |  |  |  |  |  |  |  |  | 1 |  |  |  |  | 7 | 1 |  |  |  |  |  | 2 |
| *Hypsilobarbus periyarensis* |  |  |  |  |  |  |  |  |  |  |  |  |  |  |  |  | 2 |  | 10 |  | 3 |  |  |  |  |  |  |
| *Hypsilobarbus dubius* |  | 1 |  |  |  |  |  |  |  |  |  |  |  |  |  |  | 1 |  | 15 |  |  |  |  | 1 | 3 | 3 |  |
| *Tor malabaricus* |  |  |  |  |  |  |  |  |  |  |  |  |  |  |  |  |  |  | 17 |  | 2 |  |  |  | 2 | 4 | 3 |
| *Tor kudhree* |  |  |  | 2 |  | 2 |  |  |  |  |  |  | 5 | 3 | 2 |  |  |  | 10 |  |  |  |  |  | 2 | 3 | 4 |
| *Osteochilus longidorsalis* |  |  |  |  |  |  |  |  |  |  |  |  |  |  |  |  |  |  | 5 |  |  |  |  |  |  |  | 4 |
| *Salmophasia acinaces* |  |  |  |  |  |  |  |  |  |  |  |  |  |  |  |  |  |  |  |  |  |  |  |  | 4 |  |  |
| *Barilius gatensis* | 4 |  | 4 | 18 | 11 |  | 3 | 5 |  | 5 |  | 5 | 4 | 3 |  | 2 |  | 20 | 4 |  | 5 |  |  | 20 |  |  | 10 |
| *Barilius bakeri* |  |  |  | 2 | 8 |  | 2 |  |  | 2 |  | 5 | 5 | 2 |  |  |  |  | 15 |  |  | 21 |  |  |  |  | 8 |
| *Barilius barana* |  |  |  |  |  |  | 2 | 5 |  |  |  |  |  |  |  |  |  |  |  |  |  |  |  |  |  |  |  |
| *Barilius bendelisis* |  |  |  |  |  |  |  |  |  |  |  |  |  |  |  |  |  |  |  |  |  |  | 2 |  | 3 | 3 |  |
| *Devario aequipinnatus* | 11 | 15 | 2 | 5 | 7 | 14 | 4 | 6 |  | 23 | 15 | 47 | 5 | 5 | 6 |  |  | 2 | 10 | 2 |  |  | 47 |  | 10 | 2 | 3 |
| *Rasbora daniconius* |  |  |  | 7 | 3 |  | 1 |  |  |  | 14 | 4 |  |  |  |  |  | 4 | 2 | 5 |  | 10 | 1 |  | 2 | 7 | 5 |
| *Lepiphygopsis typus* |  |  |  |  |  |  |  |  |  |  |  |  |  | 10 |  |  |  |  | 25 |  |  |  |  |  |  |  |  |
| *Garra mullya* | 20 | 2 |  | 12 | 22 | 4 | 11 | 3 |  | 7 |  |  | 8 | 7 |  |  |  | 2 | 2 |  | 2 |  |  |  | 3 | 4 | 11 |

| Species | | | | | | | | | | | | | | | | | | | | | | | |
|---|---|---|---|---|---|---|---|---|---|---|---|---|---|---|---|---|---|---|---|---|---|---|---|
| *Garra surendranathi* | | | | | | | | | 6 | | | | | | | 6 | | 3 | | | | | |
| *Garra nastuta* | | | | | | | | | | | 3 | | | | | | | | | | | | |
| *Garra periyarensis* | | | | | | | | | | | | 4 | | | | | 2 | | | | | | |
| *Garra hughi* | | | | | | | | | | | | 5 | 5 | 2 | | | | | | | | | |
| *Garra gotyola stenorynchus* | | | | | | | | | | | | | | | | | | | | 2 | 4 | | |
| *Crossochelius latius latius* | | | | | | | | | | | 1 | | | | | | | | | | | | |
| *Travancoria jonesi* | | | 1 | | 5 | | | | | 5 | 4 | 5 | 4 | 2 | | 2 | | | | | | | |
| *Nemacheilus dennisoni* | 1 | | | | | | | | | | | | 3 | | | | | | | | | | |
| *Nemacheilus guntheri* | | | 4 | 2 | 1 | 1 | | | | | | | | | | 4 | | 2 | | | | | 4 |
| *Lepidocephalus thermalis* | | 1 | | | | | | | 2 | | | | | 2 | | 3 | | | | 7 | | | |
| *Hemibagrus punctatus* | 2 | | | | | | | | | | | | | | | | | | | 1 | 3 | | |
| *Mystus cavasius* | 2 | | | | | | | | | | | | | | | | | | | 1 | 2 | 10 | |
| *Ompok bimaculatus* | | | | | | | | | 1 | | | | | | | | | | | | | | |
| *Glyptothorax housei* | | | 1 | | | | | | | | | | | | | | | | | | | | |
| *Aplocheilus lineatus* | | | | | | | | 4 | | | | | | | 7 | 5 | | | | | | | |
| *Macroganthus pancalus* | | | | | | | | | | | | | | | | | | | | 2 | | | |
| *Mastacembelus armatus* | | | | | | | | | | | | | | | | | | | | | 1 | | |
| *Chanda nama* | | | | | | 3 | | | | | | | | | | | | | | 7 | | | |
| *Peristolepis* | | | | | | 2 | 2 | | | | | | | | | | 1 | | | | | | |

| Species | 1 | 2 | 3 | 4 | 5 | 6 | 7 | 8 | 9 | 10 | 11 | 12 | 13 | 14 | 15 | 16 | 17 | 18 | 19 | 20 | 21 | 22 | 23 | 24 | 25 | 26 | 27 | 28 | 29 | 30 | 31 |
|---|---|---|---|---|---|---|---|---|---|---|---|---|---|---|---|---|---|---|---|---|---|---|---|---|---|---|---|---|---|---|---|
| *marignata* | | | | | | | | | | | | | | | | | | | | | | | | | | | | | | | |
| *Oreochromis mosambica* | 1 | | | | | | | | | | | | | | | | | | | | | | | | | | | | | | |
| *Etroplus suratensis* | | | | | | | | | | | | | | | | | | | | | | | | | | | 10 | 8 | 1 | | |
| *Etroplus maculatus* | | | | | | | | | | | | | | | | | | | | | | | | | | | 4 | | 10 | | |
| *Glossogobius guiris* | | | | | | | | | | | | | | | | | | | | | | | | | | | | | 5 | | |
| *Xenetodon cancilia* | | | | | | | | | | | | | | | | | | | | | | | | | | | | 2 | 2 | 2 | |
| *Hyporhamphus limbatus* | | | | | | | | | | | | | | | | | | | | | | | | | | | | 2 | | 1 | |

**\*Collection site number as in Table 1.**

**Table 6: Water quality of 31 study sites of six river systems**

| Sampling Locations | | pH | Conductivity (mS) | TDS (mg/L) | Resistivity (KΩ) | DO (mg/L) | Water temperature (°C) |
|---|---|---|---|---|---|---|---|
| **Moyar River System** | | | | | | | |
| Belemeenthurai | 520 | 8.4 | 59.2 | 37.7 | 16.4 | 1.3 | 24.5 |
| Gulithuraipatti | 312 | 8.4 | 57.8 | 20.37 | 24.2 | 3.5 | 23.8 |
| Kallampalayam | 300 | 7.9 | 45.2 | 28.5 | 21.9 | 2.5 | 24.1 |
| **Chalakudy River System** | | | | | | | |
| Urilikal | 3238 | 7.2 | 78.7 | 51.9 | 12.9 | 1.4 | 24.1 |
| Thellikal | 840 | 8.8 | 59.2 | 37.7 | 16.4 | 1.3 | 24.5 |
| Baghapallam | 748 | 8 | 57.8 | 38.0 | 16.8 | 2.4 | 21.7 |
| Puliyarkutti 8th bridge | 527 | 7.79 | 27.8 | 18.0 | 34.8 | 5.4 | 23.5 |
| Thunakadavu tunnel | 520 | 5.9 | 38.3 | 28.3 | 22.2 | 5.09 | 21.4 |
| Puliyarkutti 3rd bridge | 512 | 7.79 | 27.8 | 18.0 | 34.8 | 5.4 | 23.5 |
| Thunakadavu stream | 510 | 5.9 | 38.3 | 28.3 | 22.2 | 5.09 | 21.4 |
| Thenmudiparai | 510 | 8 | 45.2 | 28.5 | 21.9 | 2.5 | 24.1 |
| Orukomban range | 450 | 7.5 | 33.9 | 26.5 | 22.4 | 3.5 | 23.4 |
| Pillapara | 267 | 7.6 | 34.0 | 19.5 | 29.9 | 0.89 | 33.6 |
| Athirappalli | 202 | 7.2 | 35.2 | 47.5 | 3.97 | 0.73 | 32.7 |
| **Bhavani River System** | | | | | | | |
| Kovaikutralam falls | 560 | 7.5 | 31.3 | 20.1 | 32.3 | 3.2 | 22.5 |
| Nellithurai | 380 | 7.3 | 30.3 | 20.3 | 31.5 | 2.3 | 25.5 |
| **Periyar River System** | | | | | | | |
| Melaparai | 965 | 9 | 44.7 | 28.8 | 22.5 | 1.3 | 26.1 |
| Naduthotam | 950 | 7.5 | 46.2 | 30.4 | 20.6 | 0.7 | 25.9 |
| Ummikuppamthodu | 943 | 7.7 | 64.9 | 43.2 | 17.1 | 1.2 | 24.8 |
| Anjurily | 912 | 7.2 | 21.5 | 13.6 | 47.5 | 4.86 | 19.2 |
| Oorpannikaham | 884 | 8.3 | 50.3 | 32.3 | 20.0 | 1.2 | 24.8 |
| Sorrakottaodai | 879 | 8 | 34.2 | 21.9 | 29.5 | 1.1 | 23.1 |

| | | | | | | | |
|---|---|---|---|---|---|---|---|
| Thenkasithodu | 872 | 5.2 | 22.0 | 13.7 | 45.6 | 6.11 | 18.9 |
| Valukuparai | 869 | 7.7 | 66.9 | 43.8 | 15.1 | 0.7 | 24.8 |
| Mullaithodu | 869 | 8.1 | 78.6 | 51.4 | 12.5 | 0.9 | 24.2 |
| **Cauvery River System** | | | | | | | |
| Kadapilliyarthittu | 1137 | 9.6 | 39.1 | 26.3 | 2.58 | 0.72 | 30.5 |
| Thonanthikla | 341 | 9.2 | 39.5 | 26.3 | 2.65 | 3.11 | 30.2 |
| Belikoondu | 267 | 9.4 | 39.8 | 26.3 | 2.63 | 0.63 | 32.7 |
| Nadathittu | 262 | 9.4 | 39.8 | 26.3 | 2.63 | 0.63 | 32.7 |
| Sinnaru | 225 | 9.2 | 39.5 | 26.3 | 2.65 | 3.11 | 30.2 |
| **Nugu River System** | | | | | | | |
| Noolpuzha | 2810 | 7.32 | 85.2 | 51.7 | 11.8 | 3.62 | 23.2 |

**Table 7: Contribution of the variables (%) after Varimax rotation for Habitat characters.**

| Variables | D1 | D2 |
|---|---|---|
| Altitude | 6.940 | 45.277 |
| pH | 0.849 | 0.147 |
| Conductivity (mS) | 0.424 | 0.002 |
| TDS (ppm) | 0.568 | 0.031 |
| Resistivity (KΩ) | 0.715 | 0.075 |
| DO (mg/L) | 0.900 | 0.180 |
| Salinity (ppt) | 0.923 | 0.196 |
| Water temperature (°C) | 0.695 | 0.069 |
| Rubble & Boulders | 0.676 | 0.060 |
| Gravel | 0.740 | 0.098 |

| | | |
|---|---|---|
| Sand | 0.764 | 0.127 |
| Silt | 0.884 | 0.167 |
| Derbies | 0.828 | 0.120 |
| Bedrock | 0.714 | 0.037 |
| Stream order | 0.885 | 0.170 |
| Stream Width (m) | 0.819 | 0.155 |
| Stream Depth (m) | 0.909 | 0.196 |
| Area (m2) | 21.449 | 20.245 |
| Volume (m3) | 58.428 | 32.473 |
| Mean Velocity (m/sec) | 0.891 | 0.177 |

[Figure]

**Fig 1: Collection location of six river systems**

[Figure]

**Fig 1a: Collection location of six river systems**

[Figure]

**Fig. 2. Representation of fishes in different order among the six river systems**

[Figure]

**Fig 3: Species diversity in among 31 sites**

[Figure]

**Fig 4: Species richness among 31 sites**

[Figure]

**Fig 5: Species abundance in among 31 sites**

[Figure]

**Fig 6: Cluster dendogram shows the dissimilarity between 31 sites**

[Figure]

© Dr. A. Manimekalan & Dr. A. A. Arunkumar

**Fig. 7: Fishes collected from various water bodies of SWG**

---

## Referee Comment (RC2) · Anonymous Referee #2 · 21 Jun 2018

General comments:

The paper is an important contribution on Indian fish fauna. However, the current version severely lack the required scientific vigor, need through copy editing the entire text. I have provided substantial comments which might benefit authors to improve this paper. I have marked by comments wherever possible, highlighted repeated text, missing year (reference in the text).

Specific comments:

This paper lack coherent structure and flow. It does not mention anywhere clearly its research questions or objectives and study design. What is the sampling unit? Authors

need to cite and refer recent advances made in the field of stream ecology in India and other tropical regions on stream fish ecology. Anthropogenic pressure is very broad term, it needs to spell out clearly in the study context. Conclusion made in the paper is vague.

Analysis: Diversity analysis described is not incomplete. Fish diversity can effectively be assessed in terms of data collected on measured stream habitat variables and water quality and make subsequent inference/s. Further it needs to elaborate in greater detail in the discussion section. Too many table and figures. Common mistakes in species names. Some species names are incorrect. Very old references, missing proper reference sequence. I have highlighted in both the text as well as in the reference section.

Please also note the supplement to this comment:
https://www.earth-syst-sci-data-discuss.net/essd-2017-118/essd-2017-118-RC2-supplement.pdf

**Supplement:**

[revised manuscript text omitted]

[Figure]
 **: Species diversity in among 31 sites**

[Figure]

[Figure]

Fig 4: Species richness among 31 sites

[Figure]

: Species abundance in among 31 sites

[Figure]

[Figure]

[Figure]

Fig 6: Cluster dendrogram shows the dissimilarity between 3

[Figure]

[Figure]

[Figure]

[Figure]

**Fig. 7: Fishes collected from various water bodies of SWG**
[Figure]

---

## Author Comment (AC3) · 8 Jul 2018

Reviewer Comments and Author Responses

Comment 1: The reviewer has requested to avoid repetition in the abstract. Response 1: Based on the reviewer comments, the repetition has been removed in the abstract and the revised manuscript has been uploaded in the portal.

Comment 2: The reviewer has requested to update the introduction. Response 2: Yes the corrections has been incorporated in the manuscript as per the reviewer comment.

Comment 3: Lines 39, 40. The reviewer has requested to avoid repetition Response

3: The repetitions has been removed in the manuscript.

Comment 4: Line 72. The reviewer has requested to maintain year wise sequence i.e. Talwar & Jhingran (1991), Menon (1992) etc. Response 4: Yes the necessary corrections has been incorporated in the manuscript as per the reviewer comment.

Comment 5: Lines 81. The reviewer has requested to make the necessary corrections in the sentence. Response 5: The correction has been carried out in the manuscript.

Comment 6: Line 93. The reviewer has requested to add the SPSS software version used. Response 6: The SPSS software version has been included in the manuscript.

Comment 7: Lines 87. The reviewer has requested to change the heading from interpretive analysis to Analysis. Response 7: As per the reviewer request the heading has been changed from interpretive analysis to analysis. Comment 8: Line 89. The reviewer has suggested that diversity involve comparison across species and sites. No need to repeat again. Response 8: The necessary changes has been made in the manuscript. Comment 9: Line 95 to 100. The reviewer has advocated to remove this 2.4 section. As a researcher it is assumed that this has been done already and it is Redundant. Response 9: As per the reviewer request the necessary changes has been made in the manuscript. Comment 10: Line 103 to 114. Repetition. Either remove or briefly mention in Method section. Response 10: As per the reviewer request the details has been removed from the manuscript.

Comment 11: Lines 119 to 154. Need to rewrite this whole section. Divide this as: Briefly mention species rich sites in each studied river system. After analyzing richness and abundance with habitat and water quality parameters, write that result here. Focus on writing key results. Site wise results might be useful in discussion section. Under Result section you need to write crisp and clear in quantifiable manner.

Response 11: As per the reviewer request the whole section has been changed in the manuscript.

Comment 12: Line 158. The Reviewer has requested to check the scientific name Danio neilgherriensis.

Response 12: The scientific name for Danio neilgherriensis has been changed to Devario neilgherriensis.

Comment 13: Lines 180 to 185. Rephrase some of these sentences or remove. Response 13: As per the reviewer request the details has been removed from the manuscript.

Comment 14: Line 199. Smith (year)? Some parts can go into Introduction. But pls follow recent literature

Response 14: Year has been added in the manuscript and the latest literature has been added.

Comment 15: Line 223. No results were shown to support this argument. Response 15: Based on the reviewer comments, the results were expressed for supporting the statement.

Comment 16: Line 237 to 239. The authors are gratefully... and Provide region, state and country name.

Response 16: The region, state and country name has been provided in the manuscript.

Comment 17: The reviewer has requested to provide succinct captions for the tables and figures.

Response 17: Succinct captions has been provided for the tables and figures and the spelling for the species names has been corrected in the manuscript.

Please also note the supplement to this comment:
https://www.earth-syst-sci-data-discuss.net/essd-2017-118/essd-2017-118-AC3-

supplement.pdf